



# Measured and modelled air quality trends in Italy over the period 2003-2010

Ilaria D'Elia, Gino Briganti, Lina Vitali, Antonio Piersanti, Gaia Righini, Massimo D'Isidoro, Andrea Cappelletti, Mihaela Mircea, Mario Adani, Gabriele Zanini, Luisella Ciancarella

Laboratory of Atmospheric Pollution - Italian National Agency for New Technologies, Energy and Sustainable Economic Development - ENEA, Rome, 00123, Italy

*Correspondence to*: Ilaria D'Elia (ilaria.delia@enea.it)

**Abstract.**

Air pollution harms human health and the environment. Several regulatory efforts and different actions have been taken in

the last decades by authorities. Air quality trend analysis represents a valid tool in assessing the impact of these actions taken both at national and local levels. This paper presents for the first time the capability of the Italian national chemical transport model, AMS-MINNI, in capturing the observed concentration trends of three air pollutants, $NO_2$, inhalable particles having diameter less than 10 micrometres (PM10) and $O_3$, in Italy over the period 2003-2010. We firstly analyse the model performance finding it in line with the state of the art of regional models applications. The modelled trends result in a general

significant downward trend for the three pollutants and, in comparison with observations, the values of the simulated slopes show the same magnitude for $NO_2$ (in the range -3.0 ÷ -0.5 µg m$^{-3}$ yr$^{-1}$), while a smaller variability is detected for PM10 (-1.5 ÷ -0.5 µg m$^{-3}$ yr$^{-1}$) and $O_3$-maximum daily 8-hour average concentration (-2.0 ÷ -0.5 µg m$^{-3}$ yr$^{-1}$). As a general result, we find a good agreement between modelled and observed trends; moreover, the model allowed to extend both the spatial coverage and the statistical significance of pollutants' concentrations trends with respect to observations, in particular for $NO_2$. We

also conduct a qualitative attempt to correlate the temporal concentration trends to meteorological and emission variability. Since no clear tendency in yearly meteorological anomalies (temperature, precipitation, geopotential height) was observed for the period investigated, we focus the discussion of concentrations trends on emissions variations. We point out that, due to the complex links between precursors emissions and air pollutants concentrations, emission reductions do not always result in a corresponding decrease in atmospheric concentrations, especially for those pollutants that are formed in the

atmosphere such as $O_3$ and the major fraction of PM10. These complex phenomena are still uncertain and their understanding is of the utmost importance in planning future policies for reducing air pollution and its impacts on health and ecosystems.



## 1 Introduction

Air pollution represents one of the main environmental challenges of modern society. Numerous studies have already

demonstrated the adverse effects on health (Pope III et al., 2020; WHO, 2019; Pope III and Dockery, 2006; Cohen et al., 2017) and environment (EEA, 2020; Feng et al., 2019), as well as on climate (Watts et al., 2019; Fuzzi et al., 2015), society and economy (Lanzi and Dellink, 2019; OECD, 2016). The adverse impact on health of fine particulate matter results in premature deaths due to ischemic heart disease, strokes, lung cancer, chronic obstructive pulmonary disease and respiratory infections (Apte et al., 2018; Rajagopalan et al., 2018).

Efforts aimed at reducing air pollution have been ongoing for decades, namely in the framework of the Convention on Long-Range Transboundary Air Pollution drawn up under the United Nations Economic Commission for Europe, bringing to a general decrease of measured concentrations of air pollutants in Europe (Maas and Grennfelt, 2016). The trends in concentrations are useful to verify if and how much environmental regulations establishing limitations for pollutants' emissions, e.g. the Gothenburg protocol (UNECE, 1979) and the European Directive on National Emission Ceilings (EC,

2016), have been effective and efficient in improving air quality at national and local level. Several European studies addressed this topic, focussing on the entire continent (Colette et al., 2011, 2016, 2017a; Wilson et al., 2012; Guerreiro et al., 2014; Yan et al., 2018) and on single countries (Sicard et al, 2009; Cattani et al., 2014; Querol et al., 2014; Carnell et al., 2019; Velders et al., 2020). The studies were carried out using observed and/or modelled concentrations. The best approach should be the one which integrates both of these information. Indeed, the observed concentrations provide an actual air

quality evaluation, though at sparse locations and sometimes with poor temporal coverage, while the modelled concentrations offer a comprehensive spatial and temporal coverage, even if with intrinsic uncertainties in describing the complex processes of atmospheric chemistry and physics (Iversen, 1993).

On Europe, Colette et al. (2011) performed an assessment of nitrogen dioxide ($NO_2$), particulate matter with diameter of 10 μm or less (PM10) and ozone ($O_3$) concentrations trends over the 1998-2007 decade, using 6 regional and global chemical

transport models (CTMs). The simulated trends were evaluated against observed ones at background monitoring stations located in major anthropogenic emission hotspots. This comparison showed that the primary pollutants trends were generally well reproduced by simulations, with lower performance for $O_3$ which is a secondary pollutant produced in the atmosphere. Wilson et al. (2012) also investigated the $O_3$ trends over Europe using the CHIMERE model between 1996 and 2005. The data collected in 158 rural background stations showed that the model reproduces well the European-averaged $O_3$ trend of

the annual 5th percentiles but it is missing the positive trend in the observed 95th percentiles. Another European-wide study was conducted by Yan et al. (2018) for the period 1995-2012 using the global chemical transport model EMAC. The results showed that the model successfully captured the observed temporal variability in $O_3$ mean concentrations at EMEP background stations, as well as the contrast in the trends of 95th percentile (decreasing) and 5th percentile (increasing). Solberg et al. (2015) and Colette et al. (2017b) provided reviews of scientific papers which compare modelled to observed

trends in Europe. In the EURODELTA-Trends multi-model exercise at European scale (Colette et al., 2017a), they





investigated the period 1990-2010 with 8 chemical transport models (including the AMS-MINNI, Atmospheric Modelling System of the Italian National Integrated Model to support the international negotiation on atmospheric pollution, Mircea et al., 2014; Vitali et al., 2019). The authors showed the time variability of PM10, PM2.5 (Tsyro et al., 2017), organic aerosols and precursor gases (Ciarelli et al., 2019) and $O_3$ (Mar et al., 2016; Colette et al., 2017b). In particular, the EURODELTA-

Trends study, by analysing emissions, intercontinental inflow and meteorological variability, confirmed that the reduction of European anthropogenic emissions plays a fundamental role in the modelled net reduction of ambient air pollution.

Italy is affected by air pollution at the highest levels recorded in Europe (EEA, 2020). Despite this evidence, even if the above mentioned studies over the European area include Italy in their investigations of long-term air quality trends, few analyses focussing on the Italian territory are available. Most of the available trends analyses rely on measured

concentrations of single pollutants at single monitoring stations (Casale et al., 2000; Cristofanelli et al., 2015; Gilardoni et al., 2020) or in distinct urban areas (Cadum et al., 1999; Cattani et al., 2010; Gualtieri et al., 2014; Pozzer et al., 2019) and administrative regions (Carugno et al., 2017; Masiol et al., 2017; Lonati and Cernuschi, 2020). Some works cover the whole Po Valley, in Northern Italy, which is a well-known regional hot-spot for air pollution (Putaud et al., 2014; Bigi and Ghermandi, 2016). Currently, Cattani et al. (2014; 2018) are the only Italian-wide analyses and they are based on measured

concentrations available from the National Air Quality database (BRACE, 2013). In particular, Cattani et al. (2014) show significant reduction trends in concentrations of carbon monoxide (CO) and benzene ($C_6H_6$), linearly related with emission reductions, a large prevalence of stations measuring PM10 and $NO_2$ decreasing trends and low statistical significance in $O_3$ trends which indicates that no clear trend exists in measured ozone concentrations. So far, to the authors' knowledge, there is not a modelling study exploring concentration trends and their relations with emission changes over time covering the whole

Italian territory.

This paper evaluates the trends of three air pollutants ($NO_2$, PM10, $O_3$) in Italy, over the period 2003-2010, using the AMS-MINNI air quality model. The analysis is based on statistical methods widely used in literature, for the sake of comparability with other investigations on air quality trends. The ability of the model to reproduce the concentration trends is evaluated through the comparison with independent data available from the National Air Quality database (BRACE).

Moreover, in order to identify the potential efficacy of mitigation policies in reducing air pollution, concentrations trends were qualitatively compared to meteorological and anthropogenic emissions variability.

## 2 Data and methods

### 2.1 Air quality measurements

The air quality monitoring data considered in the present work derive from BRACE where data from regional/local
monitoring networks were collected for the formal submission to the European Environment Agency (EEA), in the framework of the reciprocal exchange of information and data from networks and individual stations measuring ambient air





pollution within the Member States (EC, 1997). BRACE fed the European database Airbase (Airbase, 2020) with data from 2002 to 2012, thus covering the period investigated in this study.

Several processing steps were applied to the raw BRACE database in order to adapt the database to model validation requirements and to verify station reported metadata, in particular concerning geographical coordinates (Piersanti et al., 2012).


In the present work, in order to analyse the concentration trends, we selected only stations covering the 100% of the investigated years with at least 75% of valid data per year. The two thresholds for time coverage were chosen according to the legal requirements on yearly time series stated in the Air Quality Directive (EC, 2008) and also widely adopted in scientific literature (Colette et al., 2011; Colette et al., 2016), for a robust analysis. The pollutants considered are $NO_2$, PM10 and $O_3$ due to their large monitoring coverage in the period of interest. Particulate matter with diameter less than 2.5 μm (PM2.5) could not be included in the analysis, as the data coverage from BRACE started in 2007 (Uccelli et al., 2017). Time resolution is given in hours (for $NO_2$ and $O_3$) and days (for PM10).


The number of the air quality monitoring stations that satisfied the chosen criteria is reported in Table 2. In Appendix S1 of the Supplementary Material (SM), Figure S1 represents the 20 Italian administrative regions and Figures S2-S4 the locations of all sites that passed the selection criteria, divided by type (background - BKG, traffic - TRA, industrial - IND) and the background ones divided by zone (rural, suburban, urban).


### 2.2 Model simulations

The air quality modelling system used for our simulations is AMS-MINNI (Mircea et al., 2014, 2016; D'Elia et al., 2009, 2018; Ciucci et al., 2016) which includes a meteorological prognostic model (RAMS), a chemical transport model (FARM), an emission processor model (EMMA) and a meteorological diagnostic processor (SURFPRO).


The three-dimensional Eulerian chemical transport model FARM (Flexible Air Quality Regional Model, http://www.farm-model.org; Gariazzo et al., 2007; Silibello et al., 2008; Kukkonen et al., 2012) describes the transport, turbulent dispersion, formation and destruction of the pollutants in the atmosphere. The mesoscale non-hydrostatic meteorological model RAMS (Regional Atmospheric Modelling System; Cotton et al., 2003) generates the required input meteorological fields. Another fundamental AMS-MINNI component is the emission processor, the Emission Manager EMMA (Arianet, 2014), which prepares the hourly gridded emissions by breaking down annual data from emission inventories in space and time. Moreover, the diagnostic module SURFPRO (Arianet, 2011), computes the Planetary Boundary Layer (PBL) scale parameters, horizontal and vertical diffusivity coefficients, deposition velocities for different chemical compounds and natural emissions, using meteorological fields in input from RAMS, orographic and land use data.



The main features of the AMS-MINNI simulation setup used to carry out the simulations are synthetized in Table 1.





**Table 1. Main features of the AMS-MINNI simulation setup.**

| Chemical Transport Model Simulation | |
|---|---|
| Model and version | FARM version 4.7 |
| Horizontal resolution | 4 km |
| Vertical layers | 16 terrain-following layers |
| Vertical extent | 10000 m |
| First layer depth | 40 m |
| Gas-phase chemistry | SAPRC99 (Carter, 2000) |
| SIA module | ISORROPIA v1.7 (Fountoukis et al., 2007) |
| SOA module | SORGAM module (Schell et al., 2001) |
| Aerosol model | AERO3 (Binkowski and Roselle, 2003) |
| In-cloud sulphate chemistry | Simplified S(IV) to S(VI) formation (Seinfeld and Pandis, 1998) |
| Boundary Conditions | Eurodelta (Colette et al., 2017a) |
| **Meteorological Simulation** | |
| Model and version | RAMS version 6.0 |
| Horizontal resolution | 12km and 4km (two way nesting) |
| Vertical | 32 levels (sigma coordinate) from 30m above ground level to lower stratosphere |
| Radiation | Chen and Cotton (1983) long/shortwave model – cloud processes considering all condensate as liquid |
| Convection | Modified Kuo scheme (Tremback,1990) |
| Lower Boundary | LEAF-2, Land Ecosystem–Atmosphere Feedback model (Walko et al., 2000) |
| Turbulence Closure | Mellor-Yamada level 2.5 scheme – ensemble–averaged TKE (Mellor and Yamada, 1982) |
| Cloud Microphysics | Bulk microphysics parameterization: cloud water, rain, pristine ice, snow, aggregates, graupel, and hail, or certain subsets of these (Walko et al., 1995) |
| Boundary conditions | GFS analyses at 0.5° horizontal resolution (https://wwdata/model-datasets/global-forcastw.ncdc.noaa.gov/data-access/model--system-gfs) |
| Data Assimilation | Nudging on pre-analysed fields |
| **Emission Processing** | |
| Anthropogenic Emissions Software and version | EMMA version 6.0 |
| Anthropogenic emissions Inventories | National Emission Inventories of Italy and neighbouring countries reported to the European Monitoring and Evaluation Programme of the UNECE Convention on Long-range Transboundary Air Pollution |
| Biogenic model e Soil-NO | MEGAN v2.04 (Guenther et al., 2006) |
| Saharan dust | None |
| Sea salt | Zhang et al. (2005) |
| Windblown dust | Vautard et al. (2005) |
| Dust traffic suspension | Amato et al. (2012); Padoan et al. (2018) |

More details about the anthropogenic emissions and the meteorological data are reported in paragraph 2.3 and 2.4, respectively.

A complete description of the standard configuration of the modelling system can be found in Vitali et al. (2019).



### 2.3 Anthropogenic emissions

Emission data used as input for AMS-MINNI simulations derive from the national emission inventories covering the period from 1990 to 2015, elaborated by ISPRA (Italian Institute for Environmental Protection and Research, Taurino et al., 2017) available in 2017. Figure 1 shows the emission variation for $SO_X$ (sulphur oxides), $NO_X$ (nitrogen oxides), PM2.5, PM10, NMVOC (non-methane volatile organic compounds) and $NH_3$ (ammonia) for the period 2003-2010 considered in the present work. The variation over the whole period, 1990-2015, by SNAP nomenclature (Selective Nomenclature for Air Pollution, see Table S1 of Appendix S2 in the SM) for the selected pollutants is reported in the SM (Appendix S2, Figs. S5-S7).

$SO_X$ emissions show the highest reduction, -58% in the period 2003-2010, followed by $NO_X$ (-29%) due to a large decrease in combustion from energy and road transport sectors, respectively. NMVOC emission reduction is driven by the road transport and solvent use sectors, while $NH_3$ emissions show a very slight decrease. PM2.5 and PM10 emissions increase from 2006 onwards due to an increase in biomass combustion in the residential sector (SNAP code 02).

The estimated emissions at national level need to be further disaggregated in space, before being assigned to the AMS-MINNI grid at 4 km spatial resolution. A provincial distribution (NUTS3 level, where NUTS stands for Nomenclature of territorial units for statistics, the hierarchical system for dividing up the territory of the European Union, https://ec.europa.eu/eurostat/web/nuts/background) is provided by ISPRA every 5 years; hence it was available for both the years 2005 and 2010. For the purposes of this work, the 2005 NUTS3 disaggregation was used for the years 2003, 2004, 2005, 2006 and 2007, while the 2010 NUTS3 disaggregation for 2008, 2009 and 2010. Finally, hourly gridded emissions on the AMS-MINNI grid were produced by means of EMMA processor.

### 2.4 Meteorological simulations

The meteorological simulations required by AMS-MINNI were elaborated making use of the RAMS model whose main features are summarized in Table 1. The hourly meteorological fields produced by RAMS, such as temperature, wind speed, relative humidity and precipitation play an important role in determining the level of air pollution concentrations. In trend analysis, it is important to establish the role of the emissions and the meteorology in influencing air pollutant concentration tendency. It is out of the scope of the present paper to attribute a relative weight to these factors in determining the analysed concentration trends, but, as a first approximation, we can consider that it could be reasonably attributed to emission trends rather than to a clear tendency in meteorology. In fact, looking at the anomalies (referred to 1981-2010 climatology) of some meteorological fields for the considered years (2003-2010) computed from NCEP/NCAR reanalyses (National Centers for Environmental Prediction/National Center for Atmospheric Research, Kalnay et al., 1996), it is worth noting that no clear tendency is shown. In Appendix S3 of the SM, yearly maps for temperature at 850hPa (T850), precipitation and 500hPa geopotential height (Z500) anomalies are reported, together with the near surface temperature trend computed from the Copernicus Climate Data Store (CDS, http://climate.copernicus.eu/climate-data-store).





Figures S8 to S10 in Appendix S3 of the SM show that the year 2004 can be considered an "average" year as it presents no anomalies for T850, Z500 and precipitation rate in the Northern area, while in the Southern area a higher precipitation rate is observed; lower values of T850 are present in 2005 while higher values are recorded in the years 2003, 2007 and 2009; lower values of Z500 are registered in the years 2005, 2009 and 2010, while higher values in the years 2003, 2006 and 2007 are recorded. A higher variability is observed for the precipitation rate with higher values in 2009 and 2010; lower values in 2006 and 2007 over Northern Italy, while almost no anomaly is observed in 2008 in Southern Italy. Moreover, no clear signal is present over Italy for near surface temperature tendency (Fig. S11): almost no trend in temperature or slightly negative in the North-Western area is observed.

### 2.5 Trend methodology

The detection and calculation of trends in measured and simulated concentrations were performed using the "openair" package (Carslaw and Ropkins, 2012), specifically designed for air pollution data analysis developed for the open source R software (version used v.3.6.1, http://www.R-project.org). The presence of a monotonic increasing or decreasing trend was estimated using the non-parametric Mann-Kendall trend test together with the Theil-Sen's method for estimating the slope of a linear trend (as a concentration variation per year) (Mann, 1945; Theil, 1950; Sen, 1968; Kendall, 1975), adopting the deseasonalisation option before time trend estimates. The calculated trends were considered as statistically significant if the significance level (i.e., the p-value of the Mann-Kendall test) is lower than 0.05 ($p < 0.05$). This method does not require assumptions about the data statistical distribution, it does not care about outliers and it has been used in several works, for example in the EMEP Task Force on Measurements and Modelling during the Eurodelta experiment (Colette et al., 2016) and in the EEA air quality trend reports (EEA, 2009; 2020). Temporal trends were calculated considering monthly averages of the pollutant concentrations at each monitoring stations. Table 2 summarizes the number of stations, grouped per type, with significant and not significant trends both for observations and simulations analyses.

**Table 2.** Number of all the stations considered in the trend analysis for the period 2003-2010 divided in background (BKG), traffic (TRA), industrial (IND) and classified as statistically significant ($p < 0.05$) for observed, simulated and both observed and simulated trends.

| | Number of stations | | | | Observations: number of stations with $p < 0.05$ | | | | Simulations: number of stations with $p < 0.05$ | | | | Number of stations where both obs and sim with $p < 0.05$ | | | |
|---|---|---|---|---|---|---|---|---|---|---|---|---|---|---|---|---|
| Pollutant | BKG | TRA | IND | *Tot* | BKG | TRA | IND | *Tot* | BKG | TRA | IND | *Tot* | BKG | TRA | IND | *Tot* |
| $NO_2$ | 36 | 33 | 4 | *73* | 26 | 19 | 2 | *47* | 32 | 33 | 3 | *68* | 22 | 19 | 1 | *42* |
| PM10 | 14 | 16 | 2 | *32* | 12 | 13 | 2 | *27* | 7 | 6 | 1 | *14* | 5 | 5 | 1 | *11* |
| $O_3$ – conc: All year | 53 | 8 | 4 | *65* | 23 | 3 | 4 | *30* | 19 | 3 | 1 | *23* | 6 | 1 | 1 | *8* |
| $O_3$ – conc: Apr-Sep | | | | | 30 | 7 | 3 | *40* | 21 | 3 | 2 | *26* | 11 | 3 | 2 | *16* |





| | | | | | | | | | | | | |
|---|---|---|---|---|---|---|---|---|---|---|---|---|
| O$_3$ – MDA8: All year | | | | 26 | 4 | 4 | *34* | 31 | 5 | 4 | *40* | 15 | 3 | 4 | *22* |
| O$_3$ – MDA8: Apr-Sep | | | | 33 | 6 | 4 | *43* | 35 | 5 | 4 | *44* | 22 | 4 | 4 | *30* |
| O$_3$ – AOT40: Apr-Sep | | | | 32 | 6 | 3 | *41* | 32 | 4 | 4 | *40* | 20 | 4 | 3 | *27* |
| O$_3$ – SOMO35 | | | | 21 | 2 | 3 | *26* | 8 | 1 | 1 | *10* | 3 | 0 | 1 | *4* |

## 3 Results and discussion

### 3.1 Model validation results

Before inspecting the AMS-MINNI capability to capture the trends of pollutants concentration, a comprehensive evaluation of the model results was carried out.

Comparisons between time series of observed and modelled values were performed on the same set of monitoring stations satisfying the selection criteria used for the trends analysis (i.e. with at least 75% of valid data per year covering all the 8 years from 2003 to 2010, see Table 2).

For all the pollutants included in the trends analysis, annual time series of daily values were used for the comparison, being this metric considered the most appropriate one for model performances assessment (Colette et al., 2011). Anyway, in 195 addition to daily values and concerning only O$_3$ evaluation, the MDA8 metric (maximum daily 8-hour average concentration), calculated for the period from April to September, was considered as well, since it turned out to be the most suitable metric for O$_3$ trends analysis within the context of this study (see Section 3.2.3).

As recommended by literature on model validation (Chang and Hanna, 2004), a comprehensive set of statistical indexes was computed in order to quantify, from different points of view, the agreement between modelled and observed values.

Therefore, three statistical metrics are presented: **M**ean **B**ias (**MB**), **R**oot **M**ean **S**quare **E**rror (**RMSE**) and the **corr**elation coefficient (**corr**); see Appendix S4 in the SM for their formulations. These indexes were chosen because they globally capture several features of model performance in terms of amplitude, phase and bias. Moreover, such indicators are frequently used in model evaluation studies (Simon et al., 2012), namely those previously cited on time trends, including Colette et al. (2011), which we consider as a reference for the present evaluation. Values of **MB**, **RMSE** and **corr** for each 205 pollutant are presented here as an average over the 8 years period and over all the available stations, being them classified according to their type (BKG, TRA, IND) and, concerning BKG stations, for zone (rural, suburban, urban). Results are shown in Fig. 2 for daily values of NO$_2$ (upper left panel), PM10 (upper right panel), O$_3$ (lower left panel) and for MDA8 of O$_3$ (lower right panel).



Overall, model performance is quite good and statistical indexes are in line with the results obtained by analogous modelling
systems (e.g. Solazzo et al., 2012; Pirovano et al., 2012; Badia and Jorba, 2015; Bessagnet et al., 2016), especially when
applied at similar spatial resolution (e.g. Chemel et al., 2010; Pay et al., 2014).

As far as $NO_2$ daily values are concerned (upper left panel of Fig. 2), AMS-MINNI turns out to perform well. More in detail,
**RMSE** and **corr** values, ranging from 10.8 to 28.6 µg m$^{-3}$ and from 0.578 to 0.689, respectively, with BKG stations scoring
best, are in line with Colette et al. (2011). According to **MB**, AMS-MINNI performs even better. Negative **MB,** between -
22.4 and -4.2 µg m$^{-3}$, are obtained for all station types, stressing a general underestimation of $NO_2$ concentration values. This
feature is commonly expected in chemical transport model applications at regional scale and it can be ascribed to the
intrinsic difficulties of regional models in capturing, at their resolution, high gradients in spatial concentration variability
(Schaap et al., 2015). This hypothesis is confirmed by the evidence that model performance (according to both **RMSE** and
**MB**) deteriorates with decreasing spatial representativeness of monitoring sites; in particular, absolute values of **MB** (i.e.
underestimations) increase passing from rural to urban environments and even more at TRA stations.

AMS-MINNI tends to underestimate PM10 daily values too, as commonly in regional models. This is highlighted by
negative values of **MB** in the upper right panel of Fig. 2. Anyway, in this case the underestimation does not seem to increase
with decreasing spatial representativeness of sites, and can be attributable to the well-known difficulties of air quality models
to take into account all the contributions to PM10 concentration (Solazzo et al., 2012; Im et al., 2015). In particular, it is
worth noting that, in the present AMS-MINNI simulations, Saharan dust contribution was not included and this could be the
main reason for the underestimation at rural sites. Anyway, compared to Colette et al. (2011), simulated PM10
concentrations are overall in agreement with observations, especially as far as **MB** and **corr** are concerned, with values
ranging from -12.8 to -3.9 µg m$^{-3}$ and from 0.453 to 0.630, respectively.

AMS-MINNI $O_3$ daily values (lower left panel of Fig. 2), are in line with the findings of Colette et al. (2011) concerning
both the general overestimation of $O_3$ concentration levels and the range of values of the statistical indexes (21.7 ÷ 25.7 µg
m$^{-3}$ for **RMSE**, 2.2 ÷ 18.6 µg m$^{-3}$ for **MB** and 0.757 ÷ 0.822 for **corr**). More in detail, considering results at background
stations, similar **RMSE** values are obtained, together with generally lower **MB** values and better correlation skills. Similarly
to $NO_2$ performance, the **MB** in reproducing $O_3$ levels changes with spatial representativeness of monitoring sites, i.e. as
$NO_2$ underestimation increases passing from rural to urban environments, $O_3$ overestimation increases, since close to $NO_2$
sources the titration process plays as $O_3$ sink (Seinfeld and Pandis, 1998).

Model performance in reproducing MDA8 of $O_3$ for the period from April to September (lower right panel of Fig. 2) is
similar to daily one, evaluated throughout the whole year, apart from the negative (albeit small) **MB** value obtained at rural
stations. With respect to daily values, MDA8 correlation skill (0.712 ÷ 0.853) is globally better, whereas no clear tendency
can be inferred concerning concentration biases: according to **MB** (-1.9 ÷ 14.8 µg m$^{-3}$), MDA8 results to be always better
(lower absolute values); according to **RMSE** (24.4 ÷ 25.1 µg m$^{-3}$), it is worse at BKG stations and slightly better at IND and
TRA ones. Nevertheless, it is worth noting that, when assessing $O_3$ performance, higher biases in concentration estimates
could be expected when using the MDA8 metric, instead of daily average, since concentration levels are higher too. Indeed,





higher MDA8 concentration values are expected if compared to daily ones, due to two reasons: i) maximum values are taken into account instead of average ones; ii) only the warm period (April-September) is considered here, when higher $O_3$ values are generally observed.

Globally, AMS-MINNI turns out to perform quite well being the results in line with the state of the art of air quality model performances, when operating at the regional scale, concerning both the values of the statistical indexes used for the comparison and the general tendency to overestimate $O_3$ and to underestimate $NO_2$ and PM10.

### 3.2 Trend analysis

From the concentration fields provided by AMS-MINNI simulations, data were extracted at each monitoring station to make a comparison between observed trends (OT) and simulated trends (ST).

In the following paragraphs, for each of the pollutants considered and for the whole set of stations described in Table 2, an analysis of observed and simulated trends is discussed examining different parameters. For each pollutant, we present:

- the overall distribution of stations with statistically significant/not significant trends, with their sign, for both observations (OT) and simulated (ST) values, in order to synthetically evaluate model performance in reproducing time trends in measured concentrations (Figs. 3, 7, 11);
- the time series of observed and simulated monthly average concentration values, detailing model performance in the considered multiannual period (Figs. 4 (a), 5 (a), 8 (a), 9 (a), 12 (a), 13 (a));
- scatter plots of observed/simulated slopes, by station type (Figs. 4 (b), 5 (b), 8 (b), 9 (b), 12 (b), 13 (b));
- the map of the simulated slopes at each grid point, in comparison with spatial distribution of observed slopes split up according to station type, in order to provide a more detailed description of the results, since observed and simulated slopes are presented according to their spatial distribution and geographical context. Moreover, simulated quantities are provided not only at monitoring sites but at every grid point of the computational domain, to fully exploit model capabilities at their best in terms of both spatial coverage and variability description (Figs. 6, 10, 14).

In Appendix S5 of the SM the observed and simulated slopes (both in terms of $\mu g\ m^{-3}\ yr^{-1}$ and $\%\ yr^{-1}$) are reported for each pollutant and for each station with a significant trend ($p<0.05$).

### 3.2.1 $NO_2$

Out of 73 monitoring stations, 47(68) show a statistically significant OT(ST) whereas 42 result to have both significant observed and simulated trends. Figure 3 shows that, concerning simulations, the 93% of the sites has a significant decreasing trend while the rest exhibits a not significant tendency. Observation based trend analysis shows a lower fraction (58%) of points with significant decreasing trends, a higher fraction (36%) of cases with not significant trends and also some cases of significant increasing trends.



Figures 4 (a) and 5 (a) show that the model better reproduces monthly values at BKG sites than at TRA and IND stations, while the intra-annual variability is well reproduced for all types of stations. This result confirms the good model performance for daily values (Fig. 2).

The scatter plots of Figs. 4 (b) and 5 (b) show an overall good agreement for BKG sites with statistically significant trends, while, as expected, performance worsen at TRA sites where the absolute values of the simulated slopes are usually lower than the observed ones.

Figure 6 shows that model simulations provide coverage and information also in some portion of the territory where observations are completely absent in the considered period, i.e. in the Southern part of Italy. Overall, at BKG stations the model better captures both the sign and the variability of the slopes, while its performances decline at TRA stations. More in general, the map of the simulated slopes shows not only a wider coverage but also a greater area of significant trends compared with observations.

### 3.2.2 PM10

The well-known underestimation of PM10 concentrations when simulated by regional models, already discussed in Section 3.1, and the poor quality of the observation network, proven also by the low number of stations fulfilling the selection criteria, greatly influenced the trend estimates. Out of 32 monitoring stations, 27(14) show a statistically significant OT(ST) while 11 have both observed and modelled significant trends. The fraction of all the sites with observed statistically significant trends, shown in Fig. 7, reaches a percentage of 84% but decreases to 44% when simulated data are taken into account. The simulated monthly mean time series illustrated in panel (a) of both Figs. 8 and 9 for BKG and TRA/IND stations, respectively, show a general underestimation of observed concentrations, with performances slightly improving from 2007 onwards. Focussing on sites where both observed and simulated trends are statistically significant, Figures 8 (b) and 9 (b) show that the model succeeds in capturing not only the sign of all the observed trends, but also the slopes in many sites, even if the absolute values are slightly underestimated, especially at the industrial site. This result is confirmed by the maps represented in Fig. 10 depicting a general agreement at most of the available monitoring stations. Although to a lesser extent if compared to NO$_2$ and O$_3$, also for PM10 the simulated statistically significant trends cover a wider area with respect to observations especially in the Northern area and in the Sardegna island where no observations are available at all. A poor coverage by both model and observations occurs in some areas of Centre and Southern Italy, where anyway the model shows larger areas of significant ST, especially for the Puglia region and Sicilia island.

### 3.2.3 O$_3$

As underlined in Lefohn et al. (2017; 2018) and Colette et al. (2017b), the choice of the O$_3$ metrics is of the utmost importance since each indicator could show a different trend. In our analysis, both effect-based indicators (AOT40 and SOMO35) and process-based indicators (MDA8) were computed and analysed. The different metrics explored are: the mean O$_3$ concentration (O$_3$ avg); the maximum daily 8-hour average concentration (MDA8); the accumulated amount of ozone





over the threshold value of 40 ppb (AOT40) calculated from April to September (Apr/Sep) and the sum of the daily maxima of 8-hour running average over 35 ppb (SOMO35) for the whole year. Concerning $O_3$ avg and MDA8 metrics, analyses were carried out on both the entire year and from April to September. Both absolute (Table 2) and relative (Fig. 11) figures of statistical significance depend on the metric. The fraction of stations with significant trend also varies between observed and

modelled trends. Annual metrics ($O_3$ avg, MDA8 and SOMO35) show a lower fraction of significant trends than the metrics calculated in Apr/Sep. For the purpose of our analysis, i.e. to show the capacity of the AMS-MINNI in capturing the air pollution trends through a comparison of observations and simulations, we preferred to focus on the MDA8 indicator calculated in the warm period (Apr/Sep), when higher $O_3$ values are generally recorded. Indeed, the MDA8 calculated in the period Apr/Sep shows the highest number of stations with significant trend among all indicators.

The fraction of stations with significant trend is comparable between observations and simulations, 66% and 68% respectively, but when looking at the sign of the trend, we found out that all significant simulated trends are decreasing, while the 39% of significant observed trends are increasing. The monthly mean shows a good agreement for BKG stations (Fig. 12 (a)) and a slight overestimation for both IND and TRA stations (Fig. 13 (a)). The scatter plots (panel (b) of both Figs. 12 and 13) show a higher variability in observed trend than simulated ones.

When looking at the spatial distribution, Figure 14 shows a wide area of significant simulated trend, ranging from -2.0 µg m$^{-3}$ yr$^{-1}$ to -0.5 µg m$^{-3}$ yr$^{-1}$ with no significant ST covering especially the North-Eastern area. The comparison with observations is particularly interesting for BKG stations, where a higher sample is available. As already pointed out, the simulated trend does not reproduce the observed trend where positive, but when significant decreasing OT are calculated the model shows a good agreement, even though with a lower variability range than observations. Moreover, there are different areas, especially

in the Central and Southern Italy, where the model shows a significant trend, whereas monitoring sites are not available at all or without significant OT.

### 3.3. Discussion

Our analysis shows the good capability of the AMS-MINNI in reproducing observed trends widening the spatial coverage of the significant modelled trends respect to observations albeit with some distinctions among the different pollutants. Although

being out of the scope of the present paper to quantitatively establish the role of emissions and meteorology in determining the analysed concentration trends, we try to carry out a preliminary qualitative attempt to compare the temporal concentrations trend to emission variations, having already observed (see Section 2.4) that there is not a clear tendency in meteorology.

$NO_X$ (i.e., nitrogen oxides that are most relevant for air pollution, namely NO and $NO_2$) are mostly emitted during fossil fuel

combustion processes, and in particular by road transport. In our analysis, the road transport sector represents almost the 50% of all the total emitted $NO_X$ (see Fig. S5 of Appendix S2 of the SM). The decrease of $NO_2$ concentrations is almost consistent with the decrease in $NO_X$ emissions, being $NO_2$ concentrations directly linked to primary emissions (Colette et al., 2011; Henschel et al., 2015) and mainly driven, in our case, by a reduction in emissions from the road transport sector.





Despite the underestimation of absolute values of background concentrations, AMS-MINNI reproduces adequately the

observed trends at national scale (Fig. 4 (b)), showing robustness in potential support to reduction policies of background pollution. On the other hand, besides underestimating concentrations at traffic stations like many state of the art CTMs, the decreasing tendency of concentrations measured at traffic stations is underestimated (Fig. 5 (b)). This indicates that the model is either misrepresenting the decrease of emissions, or producing concentrations not fully responsive to correct trends of emissions. Moreover, the spatial resolution can limit the model capacity in capturing high gradients in concentration

features, typical of the urban environment, and this may be the reason for the model failure in catching positive slopes. As an interesting example, from Fig. 6 (lower right panel) it turns out that the traffic station exhibiting the highest positive observed slope (as showed by Fig. 5 (b)) is located in Florence, Toscana region. The comparison of lower right and upper left panels of Fig. 6 points out that this traffic monitoring site, (airbase code IT0861A, see Table S2 of Appendix S5 in the SM), is located between two urban BKG sites exhibiting not significant OT. The three monitoring points are located within

about 4 km (i.e. in the same cell of the computational domain). This is a feature that the model is not able to capture; indeed, in this area simulated concentrations show no significant or decreasing trends. Something like this happens in most of the cases where positive slopes occur, being they obtained only from measurement analysis. Most of these points are very close to other monitoring sites where the opposite behaviour (negative slopes) are found; see for example the couple of BKG sites in Lombardia surrounded by other BKG sites where the opposite sign is found, or the IND site located in Eastern Liguria

very close to a TRA site showing a decreasing trend. Therefore, when designing mitigation scenarios at local urban scale, it is confirmed that a regional scale CTM like AMS-MINNI needs to be integrated with high resolution models.

Concerning PM and $O_3$, given their secondary nature, a direct link between emissions and atmospheric concentrations reductions is not expected (Guerreiro et al., 2014).

PM10 is both primarily emitted and secondarily generated in the atmosphere from chemical precursors ($NO_X$, $SO_X$, $NH_3$,

NMVOC) reactions. Therefore, observed concentrations reflect these and other contributions, like long-range transport, including Saharan dust, in quantitatively variable fractions depending on sites. The national emissions of primary PM10 (Fig. 1) are stable for the first four years (apart from 2004), then grow for two years and diminish in the last period, resulting in a final increase of 13% from 2003 to 2010. On the other hand, the emissions of all four mentioned precursors decrease at different rates. These contrasting trends in emissions could partly explain the large areas of not significant trend illustrated in

Fig. 10, whereas the areas with the higher simulated decrease correspond mainly to industrial and traffic areas, underlying the significant efforts in reducing emission from the industrial and road transport sectors. The few stations available for the comparison show a nice model skill in reproducing observed negative trends, even on TRA stations. This could be a preliminary confirmation of the fitness of AMS-MINNI for the purpose of supporting emission reduction planning, even though further evaluations of model trends are needed, especially on more recent time intervals.

$O_3$ is a totally secondary pollutant produced in the troposphere by the chemical reactions of its precursors, such as $NO_X$ and NMVOC, while $CH_4$ and CO become more important at a wider scale (Guerreiro et al., 2014).



The number of cited studies focussing on O$_3$ indicates how critical this pollutant is when exploring relations between emission and concentration time trends, given the complex generation chain, showing sometimes a discrepancy between the emission decrease of O$_3$ precursors and the variation of O$_3$ concentrations (Colette et al., 2011; Guerreiro et al., 2014; Querol

et al., 2014). This is even more important in the particularly susceptible to ozone-related impacts Mediterranean area (De Marco et al., 2019) whose climatological conditions are more favourable for its formation (Guerreiro et al., 2014). The national emissions of the main ozone precursors follow a similar descending gradient in the 8 years considered, so not modifying the national scale ratio between the two, which is the main driver of the chemical equilibrium in the generation of O$_3$ (Seinfeld and Pandis, 1998; Sillman, 1999). This could partly explain (Fig. 14) why the model gives not significant or

close to zero trends in Northern Italy, especially in the Po Valley, a well-known air pollution hot spot densely populated with high anthropogenic emissions. In the same region, where most of the monitoring stations are concentrated, different behaviours of OT are observed in some scattered stations: negative slopes (especially in the Western part), not significant trends and positive slopes. These stations are mainly located in complex orographic contexts or near to the coastline, where the transport of O$_3$ from the sea, caused by sea breeze circulation (Monteiro et al., 2016), together with precursors' emissions

by nearby harbours, could be additional causes of local peculiar features. Similar findings were already noticed in literature, as for example in Guerreiro et al. (2014) or in Colette et al. (2011), who noticed in particular that different models had different behaviours. On a national level, Cattani et al. (2014), focussed on observations all over the Italian territory in the period 2003-2012, showed that it is not possible to estimate a general significant statistical trend (even if on a different reference metric, i.e. SOMO0 calculated in the half summer period), regardless of the type or the area of the stations, and

that there are discrepancies in significant trend among adjacent stations. Moreover, as already mentioned, the choice of the O$_3$ metrics can influence the trend estimate. Overall, in our analysis, the AMS-MINNI underestimates the absolute value of the descending OT at background stations. This result is driven by North-Western monitoring locations, where further work is needed to analyse the quality of local emission values and external contributions to ozone concentrations.

### 4 Conclusions

The present work describes the first assessment of the capability of the Italian chemical transport model AMS-MINNI in capturing the trends of three pollutants, namely NO$_2$, PM10 and O$_3$. The analysis on O$_3$ was carried out using different metrics, both for observations and simulations. We firstly conducted a thorough analysis of the model skill considering some statistical score parameters most commonly used in literature. This analysis confirms that the model performance is in line with the state of the art of regional models applications. Statistical indicators turned out to be as good as other CTMs in

literature and the same behaviour of most of the regional models was observed concerning the general tendency to overestimate O$_3$ and to underestimate NO$_2$ and PM10. The trend evaluation was performed using the non-parametric Mann-Kendall trend test together with the Theil-Sen's method for the estimation of the slopes and an in-depth comparison between observed and AMS-MINNI modelled trends was carried out. Comparing the sign of modelled and observed trends we found





out a good agreement for almost all sites. Our main result is a general downward simulated trend for the three pollutants. With respect to observations, modelled slopes show the same magnitude for $NO_2$ (in the range $-3.0 \div -0.5$ µg m$^{-3}$ yr$^{-1}$), while a smaller variability is detected for PM10 ($-1.5 \div -0.5$ µg m$^{-3}$ yr$^{-1}$) and $O_3$-MDA8 ($-2.0 \div -0.5$ µg m$^{-3}$ yr$^{-1}$). Concerning PM10, the reason for this discrepancy could be attributed to the well-known underestimation of PM10 modelled concentrations. $O_3$ results could be influenced by the poor quality of the monitoring network in the period we considered, together with the well-documented difficulties of models in catching $O_3$ concentration trends, given its non-linear

dependence on precursors emissions.

Model capabilities in terms of both spatial coverage and variability description were illustrated by the maps generated for all the three pollutants, showing a wider significant area for simulated trends compared to observed ones, with a higher coverage for $NO_2$ and $O_3$-MDA8 and a lower one for PM10. For almost all Northern Italy and for all pollutants, it is possible to estimate an area with significant simulated trends. Even for Southern Italy, where in general a low coverage of PM10

significant modelled trends is obtained, it is possible to catch some areas with simulated significant trends where no observed data are present at all. More in detail, it is worth noting that in the major islands, Sardegna and Sicilia, the simulated trends give useful information, filling the gap due to the absence of significant observed trends.

Moreover, a qualitative comparison between the temporal concentration trends and the meteorological and emission variations was carried out too. Since we do not observe a clear tendency in meteorology anomalies, concentrations trends

were discussed in connection with emissions variations. Indeed, it was pointed out that, due to the complex links between precursors emissions and air pollutants concentrations, emission reductions do not always result in a corresponding decrease in atmospheric concentrations, especially for secondary pollutants like PM10 and $O_3$. Studies on air pollutant trends are relevant to evaluate the impact of the actions taken to reduce emissions in different environmental policies both at national and local levels. Our analysis demonstrates the good agreement between modelled and observed trends and the added value

of the model in widening both the coverage and the significance of air concentration trends with respect to observations. Model performance turned out to be better for $NO_2$, while for the others, especially $O_3$, the issue is more challenging. Moreover, the capability to interpret past air quality trends is fundamental in understanding the efficacy of already applied air quality policies and measures and in planning further actions. As demonstrated, the understanding of complex interactions is still uncertain and represents a gap to be filled since it is of the utmost importance in planning future policies

aimed at reducing air pollution and its impacts on health and ecosystems.



**Appendix A: list of acronyms**

AMS = Atmospheric Modelling System

BKG = Background

BRACE = Banca Dati e Metadati di Qualità dell'aria (National Air Quality database)

corr = correlation coefficient

CTM = Chemical Transport Model

ECMWF = European Centre For Medium-Range Weather Forecast

EEA = European Environmental Agency

EMAC = ECHAM/MESSy Atmospheric Chemistry

EMEP = European Monitoring and Evaluation Programme

FARM = Flexible Air Quality Regional Model

IND = Industrial

ISPRA = Istituto Superiore per la Protezione e Ricerca Ambientale (Italian Institute for Environmental Protection and Research)

MB = Mean Bias

MDA8 = Maximum Daily 8-hour Average

MINNI = Modello Integrato Nazionale a supporto della Negoziazione Internazionale sui temi dell□Inquinamento atmosferico (Italian National Integrated Model to support the international negotiation on atmospheric pollution)

$NO_2$ = nitrogen dioxide

NUTS = Nomenclature of territorial units for statistics

$O_3$ = ozone

OECD = Organization for Economic Co-operation and Development

OT = Observed Trends

PBL = Planetary Boundary Layer

PM10 = particulate matter with diameter of 10 µm or less

RAMS = Regional Atmospheric Modelling System

RMSE = Root Mean Square Error

SIA = Secondary Inorganic Aerosol

SM = Supplementary Material

SNAP = Selective Nomenclature for Air Pollution

SOA = Secondary Organic Aerosol

ST = Simulated Trends

SURFPRO = SURFace-atmosphere interface PROcessor

TRA = Traffic

WHO = World Health Organization

WMO = World Meteorological Organization

**Code and data availability**

The meteorological model RAMS v6.0 is freely available at http://www.atmet.com/software/rams_soft.shtml. The chemical transport model FARM v4.7.0 is freely available at https://hpc-forge.cineca.it upon request to ARIANET s.r.l. (http://www.aria-net.it). The emission software emma6 is available on charge upon request to ARIANET s.r.l. All the codes can be provided confidentially for the editor and reviewers in order to enable peer review. All the modelled data (gridded

emissions, meteorological and concentrations fields at 4 km resolution) and the trend analysis elaborations are available upon request to the authors. Observation data are publicly available from the BRACE website (http://www.brace.sinanet.apat.it/web/struttura.html).

**Authors contribution:**

ID, LV, GR, AP put original effort into: conceptualization, methodology, study and interpretation of the trend analysis. MD,

GB, AC performed the model simulations; ID and GB carried out the data processing; ID, LV, GR, AP, MD wrote the original draft, including visualization, and performed the review and editing; AC, GB, MA, MM, GZ, LC contributed to the writing review and editing; GZ and LC accomplished the acquisition of funds.

**Competing interests:**

The authors declare that they have no conflict of interest.

**Acknowledgements**

The computing resources and the related technical support used for the AMS-MINNI simulations were provided by CRESCO/ENEAGRID High Performance Computing infrastructure and its staff (Iannone et al., 2019). The infrastructure is funded by ENEA, the Italian National Agency for New Technologies, Energy and Sustainable Economic Development and by Italian and European research programmes (http://www.cresco.enea.it/english). All the simulations were performed under

the contract agreement N. 1400519908 between ENEA and ENEL funded by ENEL. We would like to express our thanks to Giorgio Cattani (ISPRA) for the support in gathering observation data.



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





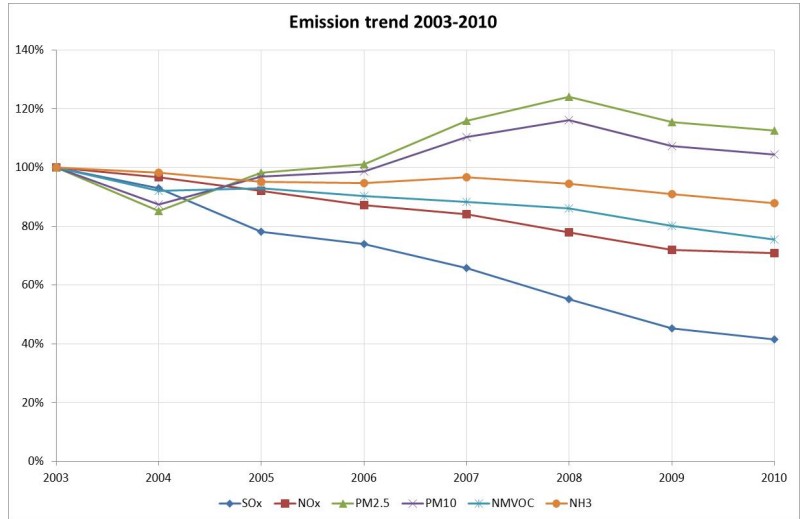

**Figure 1: Italian anthropogenic emissions, as a percentage relative to 2003, from 2003 to 2010, elaborated from ISPRA emission data set described in Taurino et al. (2017).**




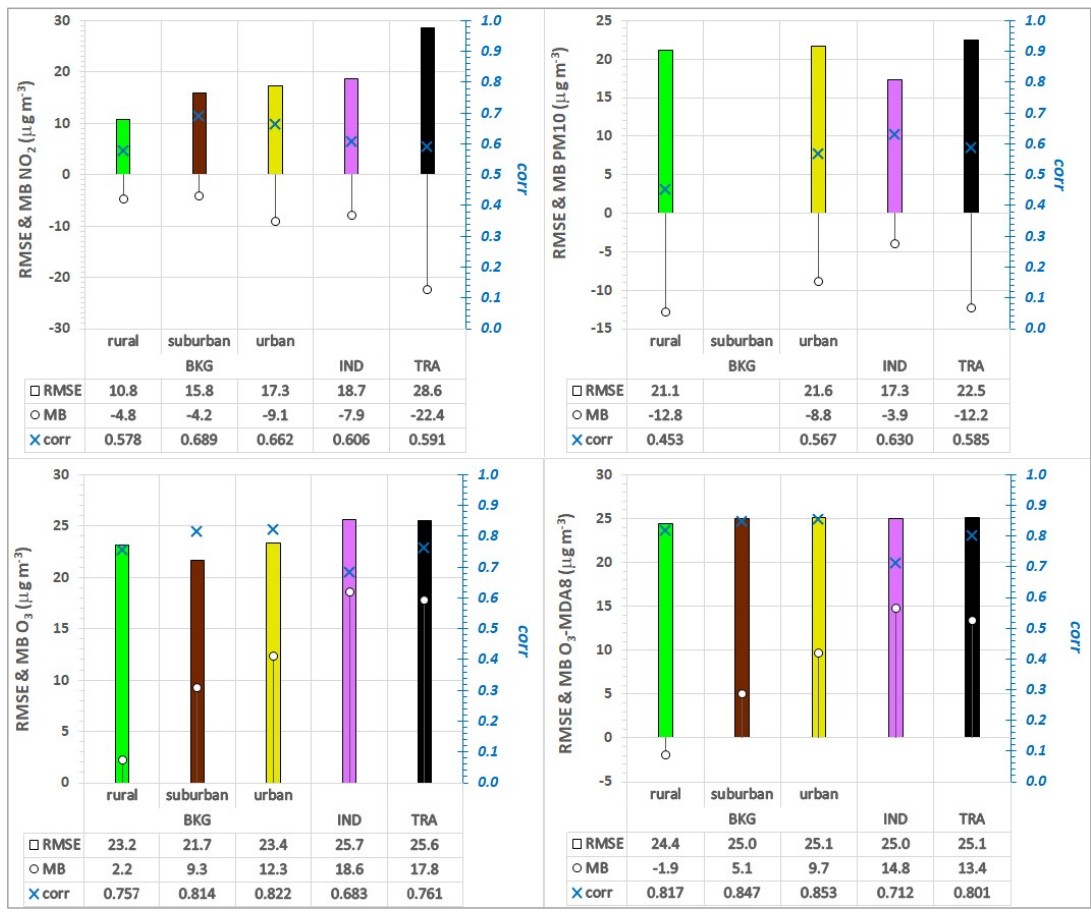

**Figure 2: Summary of model performance evaluated at all valid Italian monitoring stations during 2003-2010. The statistical scores are based on annual time series of daily average values of NO$_2$ (upper left panel), PM10 (upper right panel) and O$_3$ (lower left panel) and on MDA8 of O$_3$, calculated in the period Apr-Sep (lower right panel).**






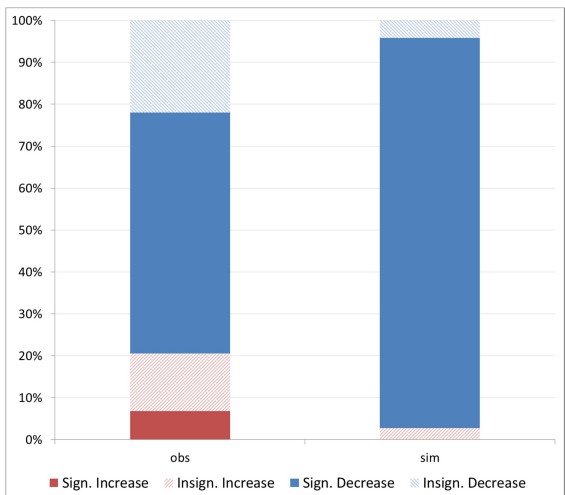

**Figure 3: Percentage of sites where statistically significant upward trends (dark red), not significant upward trends (dashed dark red), significant downward trends (dark blue) and not significant downward trends (dashed dark blue) were obtained for $NO_2$ observations and simulated data.**


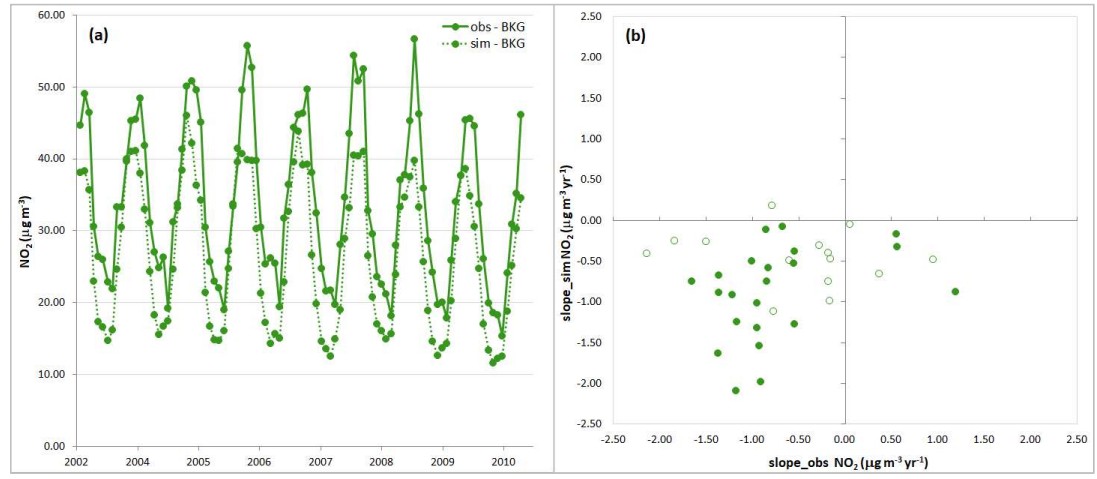

**Figure 4: (a) Observed (solid line) and simulated (dash line) monthly means of $NO_2$ concentrations (in µg m$^{-3}$) for all the background monitoring stations. (b) Scatter plot of observed and simulated slopes (in µg m$^{-3}$ yr$^{-1}$) at each individual stations. Sites where significant slopes are estimated for both observations and simulated data are indicated with a filled symbol.**






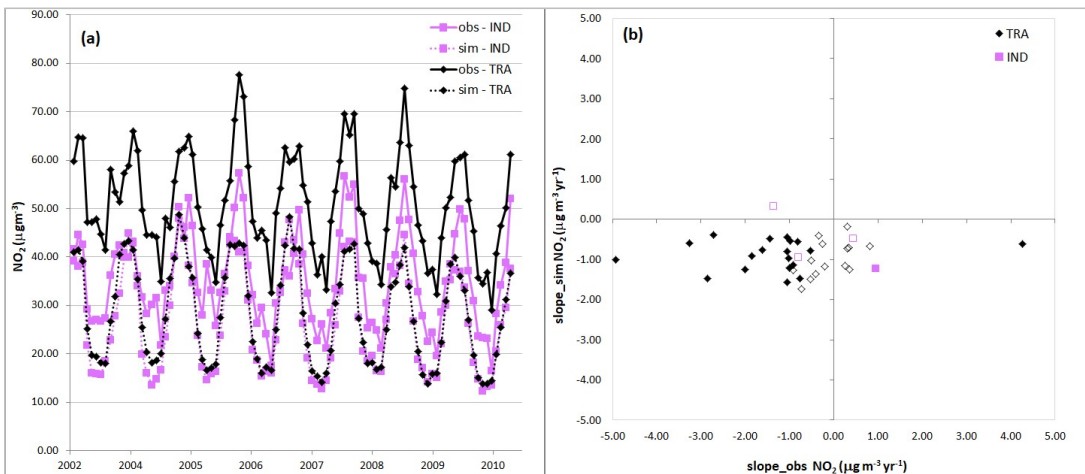

**Figure 5: (a)** Observed (solid line) and simulated (dash line) monthly means of $NO_2$ concentrations (in µg m$^{-3}$) for all the traffic (black diamond) and industrial (lilac square) monitoring stations. **(b)** Scatter plot of observed and simulated slopes (in µg m$^{-3}$ yr$^{-1}$) at each individual stations. Sites where significant slopes are estimated for both observations and simulated data are indicated with a filled symbol.



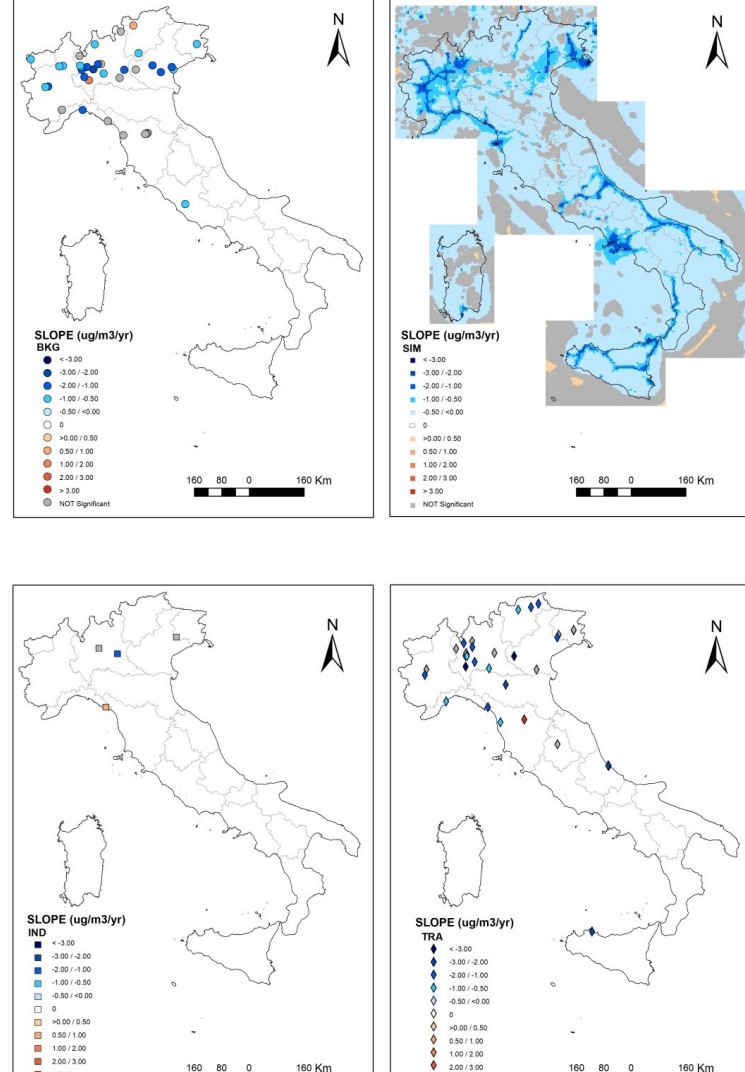

**Figure 6: Slopes of NO₂ (µg m⁻³ yr⁻¹) observed at background (BKG – upper left panel), industrial (IND – lower left panel) and traffic (TRA – lower right panel) stations and simulated (upper right panel) slopes at each grid point. The grey symbols refer to not significant trends for both the observations and the simulated data.**




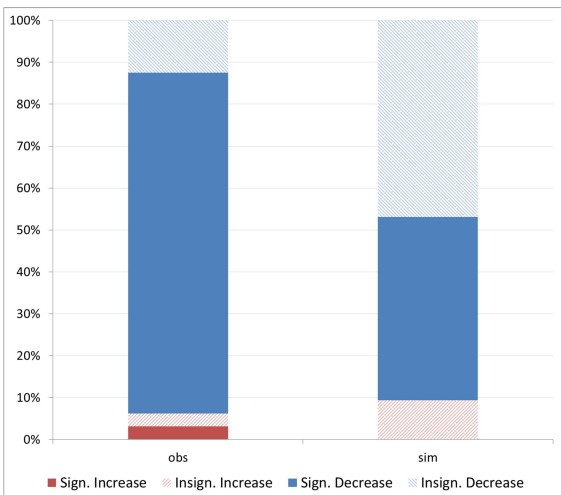

**Figure 7: Percentage of sites where statistically significant upward trends (dark red), not significant upward trends (dashed dark red), significant downward trends (dark blue) and not significant downward trends (dashed dark blue) were obtained for PM10 observations and simulated data.**


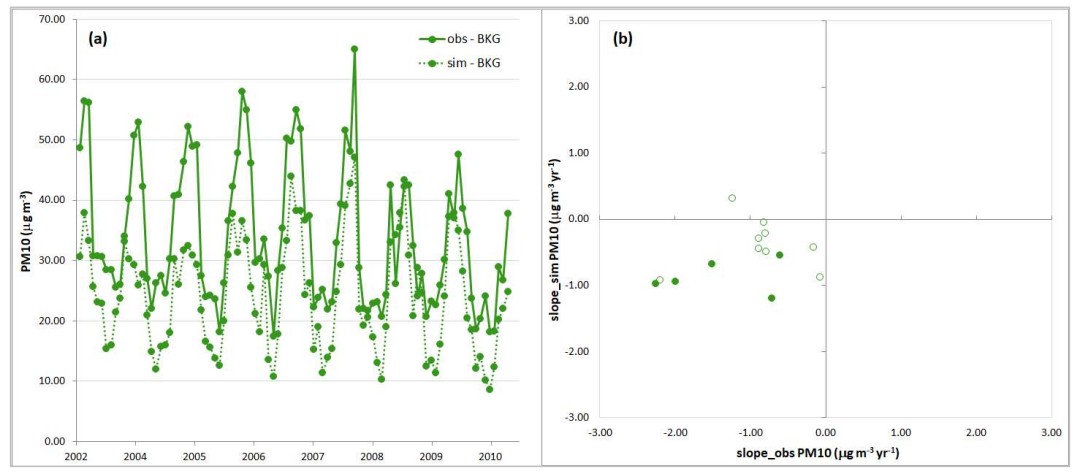

**Figure 8: (a) Observed (solid line) and simulated (dash line) monthly means of PM10 concentrations (in µg m⁻³) for all the background monitoring stations. (b) Scatter plot of observed and simulated slopes (in µg m⁻³ yr⁻¹) at each individual stations. Sites where significant slopes are estimated for both observations and simulated data are indicated with a filled symbol.**




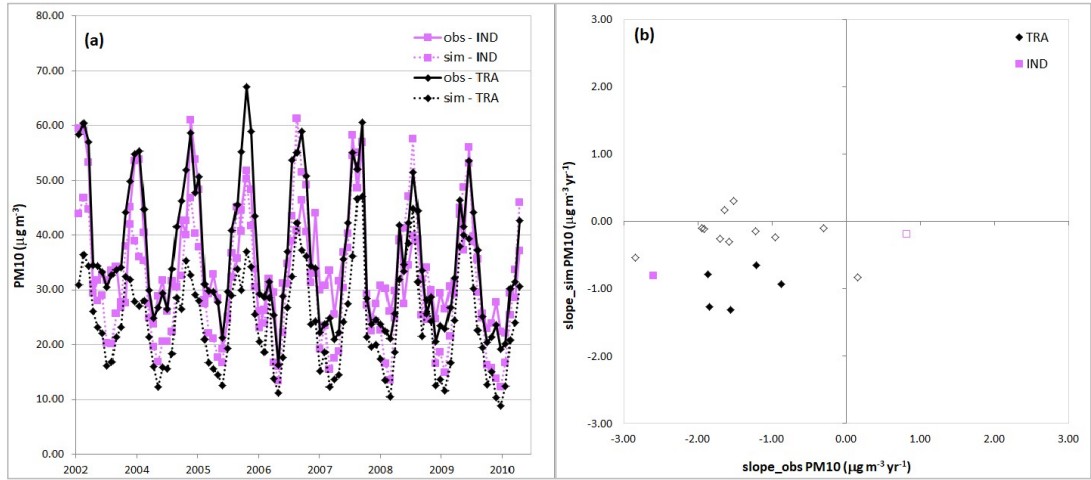

**Figure 9: (a) Observed (solid line) and simulated (dash line) monthly means of PM10 concentrations (in µg m⁻³) for all the traffic (black diamond) and industrial (lilac square) monitoring stations. (b) Scatter plot of observed and simulated slopes (in µg m⁻³ yr⁻¹) at each individual stations. Sites where significant slopes are estimated for both observations and simulated data are indicated with a filled symbol.**




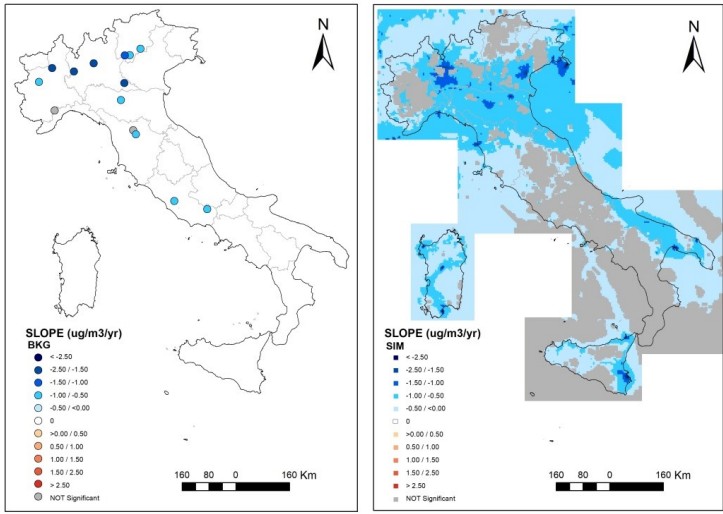

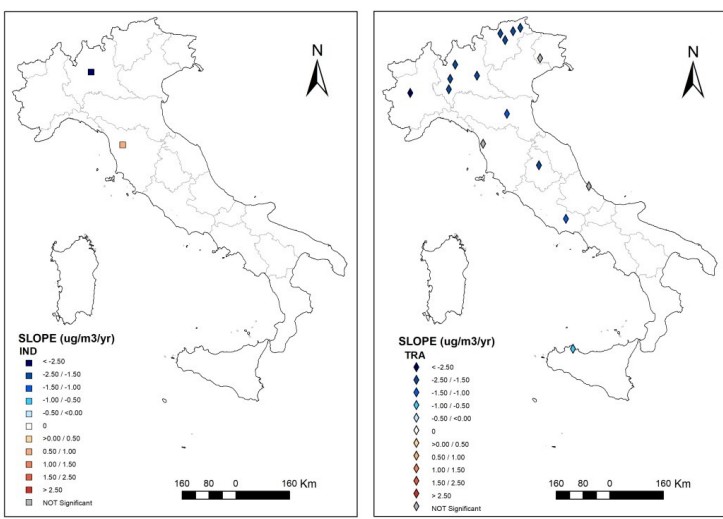


**Figure 10: Slopes of PM10 (µg m⁻³ yr⁻¹) observed at background (BKG – upper left panel), industrial (IND – lower left panel) and traffic (TRA – lower right panel) stations and simulated (upper right panel) slopes at each grid point. The grey symbols refer to not significant trends for both the observations and the simulated data.**





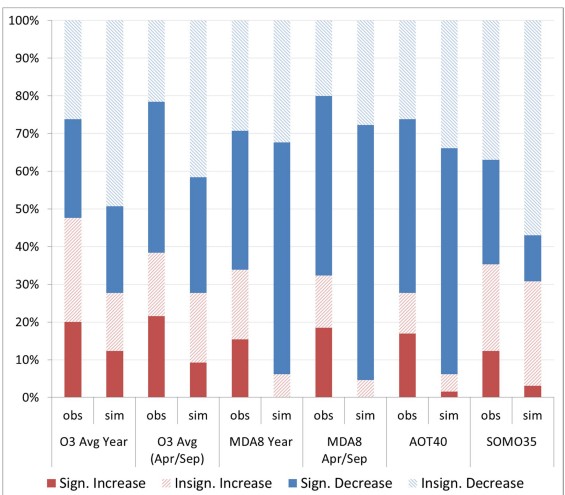

**Figure 11: Percentage of sites where statistically significant upward trends (dark red), not significant upward trends (dashed dark red), significant downward trends (dark blue) and not significant downward trends (dashed dark blue) were obtained for different $O_3$ observed and simulated metrics.**

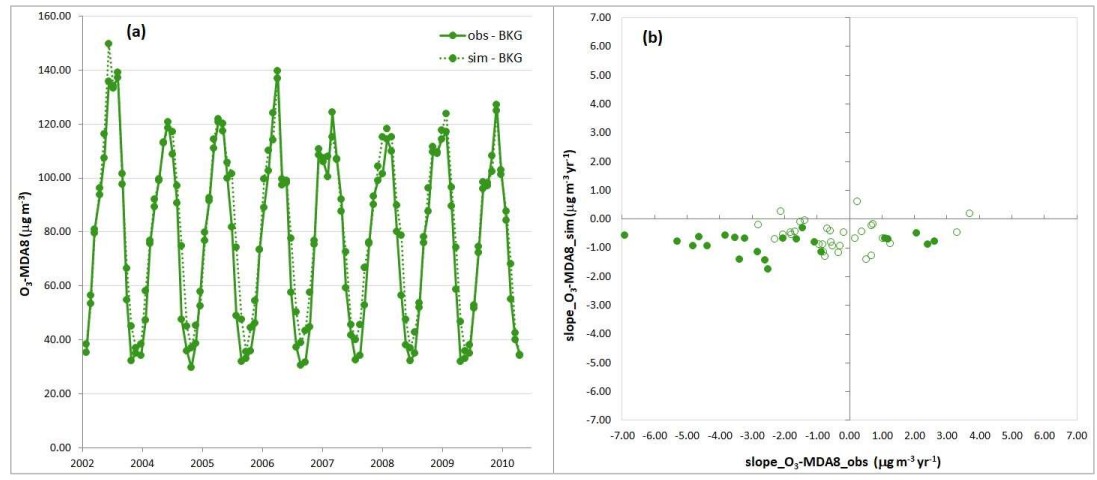

**Figure 12: (a) Observed (solid line) and simulated (dash line) monthly means of $O_3$-MDA8 concentrations (in $\mu g\ m^{-3}$) for all the background monitoring stations. (b) Scatter plot of observed and simulated slopes (in $\mu g\ m^{-3}\ yr^{-1}$) for $O_3$-MDA8 in the period April/September at each individual stations. Sites where significant slopes are estimated for both observations and simulated data are indicated with a filled symbol.**





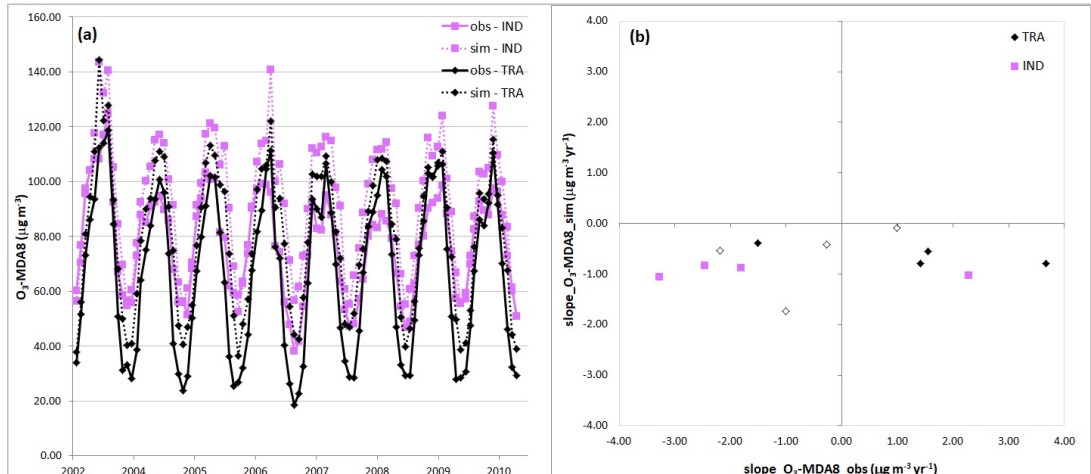

**Figure 13: (a) Observed (solid line) and simulated (dash line) monthly means of O$_3$-MDA8 concentrations (in μg m$^{-3}$) for all the traffic (black diamond) and industrial (lilac square) monitoring stations. (b) Scatter plot of observed and simulated slopes (in μg m$^{-3}$ yr$^{-1}$) for MDA8 in the period April/September at each individual stations. Sites where significant slopes are estimated for both observations and simulated data are indicated with a filled symbol.**






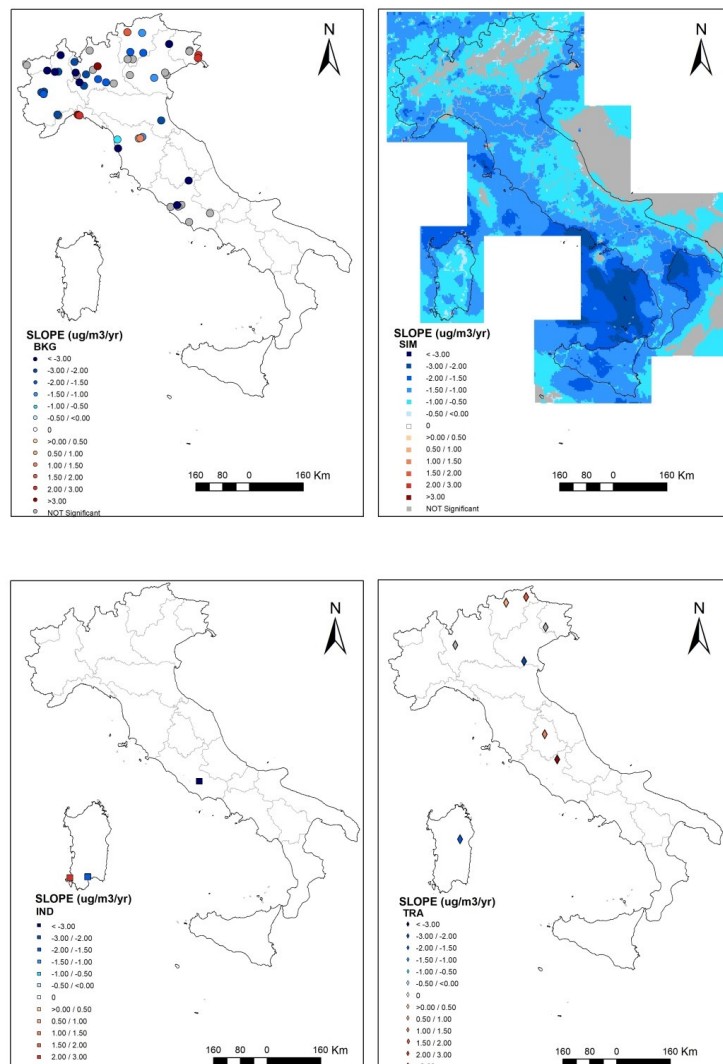

**Figure 14: Slopes of O₃-MDA8 (μg m⁻³ yr⁻¹) observed at background (BKG – upper left panel), industrial (IND – lower left panel) and traffic (TRA – lower right panel) stations and simulated (upper right panel) slopes at each grid point. The grey symbols refer to not significant trends for both the observations and the simulated data.**