# Peer review of "Measured and modelled air quality trends in Italy over the period 2003-2010"

_Atmospheric Chemistry and Physics, 2020_

## Author Comment (AC1)

We would like to thank the reviewer for the constructive comments and valuable suggestions. We address below all the issues (blue text) and specify where the manuscript has been updated accordingly.

**Anonymous Referee #2**

**RC1: 'Comment on acp-2020-1058'**

This manuscript analyses and discusses both observed and simulated trends of three air pollutants (PM10, $NO_2$ and ozone) in Italy over an 8-year period. I have somewhat mixed feelings on the manuscript. On one hand, the paper is clearly written and well organized, and it appears to be scientifically sound. On the other hand, the paper is not particularly original, and it is unlikely to attract a wide scientific interest. I have three major points that the authors should address before I can recommend accepting this paper for publication.

First, since this paper deals with air pollution trends, I am concerned with the rather short time period (2003-2010) covered in the analysis. This is a very short time periods considering potentially large year-to-year variability in meteorological conditions affecting air pollutant concentrations. The authors mention (section 2.3) that the national emission inventory covers the period 1990 to 2015, and it is difficult to see what would limit running the model simulations for a longer period as well. Concerning emission data, I understand that there might be fewer stations with sufficient data coverage prior to 2003, but why not to look at the trends until 2015?

*We agree with the reviewer that the period investigated could be longer but this does not influence the value of the results shown here. Several studies have already published observed trend studies on periods shorter than 10 years and we list here few examples:*

- *Zhai et al., 2019, Fine particulate matter (PM2.5) trends in China, 2013–2018: separating contributions from anthropogenic emissions and meteorology, ACP, 19, 11031-11041 where the authors affirm that "Trend analyses use only those sites with at least 70 % data coverage for each of the 6 years from 2013 to 2018";*
- *Dufour et al., 2018. Lower tropospheric ozone over the North China Plain: variability and trends revealed by IASI satellite observations for 2008–2016, ACP, 18, 16439–16459.*
- *Sheng et al., 2018. 2010–2016 methane trends over Canada, the United States, and Mexico observed by the GOSAT satellite: contributions from different source sectors, ACP,18, 12257–12267, where it is said that "We use 7 years (2010–2016) of methane column observations from the Greenhouse Gases Observing Satellite (GOSAT) to examine trends in atmospheric methane concentrations over North America and infer trends in emissions".*

*Nevertheless, we are aware that an analysis of longer time series would strengthen our findings. To overcome the issue of the number of year availability, in applying the chosen method a more stringent criterion was adopted with respect to other studies, i.e. only stations with valid data covering the 100% of the investigated years were taken into account (for a total of 8 years), whereas a less stringent criterion is generally adopted in other studies (as for example, 75% is set in Colette et al., ACP, (2011), corresponding in some cases to 8 years).*

*In addition, the choice of the period to investigate in this first study of observed and modelled trends over Italy was also determined by the availability of coherent model results that have the same model setup for the years 2003 to 2010. More specifically, in the following years, AMS-MINNI simulations adopted a different setup (spatial domain, chemical mechanism, boundary conditions), that clearly affects time series homogeneity.*

*We have added the following text in the Data and Methods section to explain why this period was chosen (Line 103-111 of the revised version of the manuscript):*

*"The threshold of 100% of the investigated years is a more stringent criterion with respect to other studies, generally adopting a less stringent criterion (e.g. 75% is set in Colette et al. (2011), corresponding in same cases to 8 years). Our choice guarantees that the trend analysis is always based on an 8-year period, which can be considered quite robust. Indeed, several studies are available in literature, presenting trend analysis over similar or shorter periods (Zhai et al., 2019; Dufour et al., 2018; Sheng et al., 2018). Of course, data covering a longer period would strengthen our findings. Anyway, in this first study over Italy, the choice of the period to investigate was determined by the availability of coherent model results that have the same model setup for the years 2003 to 2010. More specifically, in the following years, AMS-MINNI simulations adopted a different setup (spatial domain, chemical mechanism, boundary conditions), that clearly affects time series homogeneity."*

Second, the authors should explain more explicitly what is the scientific purpose of this paper. Evaluating the trends of air pollutants, as stated on line 81, does not really mean anything. After reading the paper, it seems that this paper is mostly about model evaluation, more specifically about the capability of the model AMS-MINNI in simulating air pollutant trends in Italy. It remains somewhat unclear whether the authors want to say anything about the actual air pollution trends over the considered time periods, based on either modeled or simulated data or some combination of these two.

*The paper is focused on the evaluation of the capability of the model to capture measured trends but it also point out the strengths and the weaknesses of CTMs in reproducing simultaneously the trends of three pollutants of major concern for human health in Italy and Europe. We think that presenting how reliable is a model to build multi-pollutant time trends is an important prove of CTM capabilities in assessing air quality and supporting air quality plans, especially for models regularly used in national regulatory assessments, as requested by the Air Quality and National Emission Ceilings directives but also for other scientific studies. Indeed, our analysis demonstrates the good agreement between modelled and observed trends and the added value of the model outcomes to provide coverage and information also in parts of the territory where observations are completely absent or observed time trends turn out to be statistically not significant. We have added the following text to better clarify this point:*

*Introduction, Line 82-85 of the revised version of the manuscript:*

*"The evaluation of CTM capabilities to reproduce the trends of pollutants increases the reliability of their application in assessing air quality and supporting air quality plans, especially for models regularly used in national regulatory assessments, as requested by Air Quality (EC, 2008) and National Emission Ceilings (EC, 2016) directives but also for other scientific studies."*

*Conclusions, Line 429-431 of the revised version of the manuscript:*

*"The evaluation of the AMS-MINNI capability to reproduce the trends of pollutants increases the reliability of its application in assessing air quality and supporting air quality plans, especially for its use in national regulatory assessments."*

Third, in order to attract more readers, the authors should at least shortly discuss the applicability of this paper to other part of the world, and in simulating the temporal variability of other important air pollutant than the three ones considered in this work.

*The methodology applied in the present paper to obtain a synoptic picture of the concerned air quality trends is widely recognized in literature (as underlined throughout the manuscript) and can be extended to other part of the world.*

*The results shown here may be a reference for other studies in complex geographical conditions such as the Italian territory, that represents an interesting environmental framework due to its complex orography resulting in peculiar meteorological conditions, the great variety of natural and anthropogenic contexts, and the presence of the Po Valley, a well-known air pollution hot spot. For the first time on the entire Italy, different $O_3$ metrics were considered in analysing both observed and modelled trends, being $O_3$ a particular significant pollutant in the Mediterranean area.*

*We have added this short discussion in the Conclusion section underlying also the importance of analysing other air pollutants (Line 438-442 of the revised version of the manuscript)*

*"The present analysis may be applied to other pollutants, especially substances of potential concerns for health (e.g. PM2.5). Moreover, it can be considered a reference for other studies in complex geographical conditions such as the Italian territory, that represents an interesting environmental framework, due to its complex orography, resulting in peculiar meteorological conditions, the great variety of natural and anthropogenic contexts, and the presence of the Po Valley, a well-known air pollution hot spot."*

---

## Author Comment (AC2)

We would like to thank the reviewer for the very useful and constructive feedback, which helped improving the quality of the paper.

We address below each specific issue (blue text) and the manuscript has been updated accordingly.

**Anonymous Referee #3**

**RC2: 'Comment on acp-2020-1058'**

This paper assesses the ability of the AMS-MINNI chemistry transport model to estimate air pollutant concentrations and trends over the period 2003-2010. In my opinion these types of assessment very important for justifying the use of these types of models for estimating the possible effects of national and international legislation that influence pollutant emissions and their impacts on air quality. As the authors point out, this type of assessment is not new and cite various studies with similar assessments, both national and European. However, there is a need to carry out such an assessment for Italy, and more specifically for the AMS-MINNI model to justify the use of this model in national regulatory assessments. After an introduction describing the problems of air quality and the use of models to understand and contribute to solutions to these problems, the authors describe the air quality measurements, model simulations and statistical methods used in the assessment. The results of the assessment are presented in two sections, the first of which evaluates the ability of the AMS-MINNI model to reproduce the observed concentrations and the second section evaluates the ability of the model to reproduce the trends in the observed values for the period 2003-2010. The authors conclude that the model is fit for purpose with regards to its ability to estimate atmospheric concentrations of NO2, O3 and PM10 although some discrepancies are found, such as an underestimation of NO2 and PM10 concentrations and an overestimation of O3 concentrations at urban and suburban sites. With regards to the trends the authors classify the model performance for estimating the trends as good, although again with some discrepancies, such as an underestimation of NO2 trends for traffic stations and an underestimation of O3 trends at nearly all sites, including the estimation of trends in the opposite direction to those observed. In summary, the assessment is not novel but does provide a valuable contribution to the body of knowledge regarding the ability of air quality models to reproduce observed concentrations and trends of air pollutants as well as contributes to the work justifies the use of AMS-MINNI model in regulatory assessments. However there are some specific points that I highlight below, which I think the authors should address before accepting the manuscript for publication.

The most important of these regards the conclusions by the authors that the model's ability to reproduce the observed concentrations is in line with the "state of the art". This may be the case but it is not clear to the reader. With the way the analysis is presented, the reader does not get an impression of whether the model performance is good or not, other than subjective statements in the text like "model performance is quite good". I suggest that the performance indicator values are explicitly compared (e.g. in a table) with the values of Colette et al. (2011), used as a reference for this evaluation, and other studies to show that the results are in line with the state of the art.

*Thank you for the useful suggestion. A table was added in SM, for an explicit comparison of our performances with what obtained by Colette et al. (2011). Indeed, Colette et al. (2011) is a reference paper for our evaluation, for several reasons:*

*i) "a short model evaluation" (as authors defined it at the beginning of their Section 4) is presented there as a preliminary step for the trends analysis, which is the main goal of their work;*

*ii) the same set of three statistical indices (i.e. RMSE, MB, corr) is discussed;*

*iii) the same metrics of concentration values (i.e. daily average values) are used for the validation.*

*We agree that the addition of the new Table improves Section 3.1, since it makes the comparison of MINNI performances with Colette et al. (2011) easier to follow (throughout the Section).*

*Moreover, sentences like "performance is quite good" or "AMS-MINNI turns out to perform well" were removed and all comments are now based on the direct comparison of values. The sentence "Globally, **AMS-MINNI performs quite well**, with the results being in line with the performances of state of the art of air quality models....." (according to the new proposed version) is present only at the end on the Section, after the comments of the results, so it turns out to be well-founded.*

Finally, although the level of English is suitable for understanding the manuscript's content, it could definitely be improved. I have provided below a comprehensive list of suggested changes that could improve the readability of the manuscript.

*We want to thank this anonymous reviewer for his/her suggestions and for having provided a very comprehensive and precise list of suggested changes that we have really appreciated and taken into account.*

**Specific comments**:

Line 1: Why was this period chosen for the analysis. This should be somewhere in the Introduction or Methods section

*We accept the suggestion adding the following sentence in the Methods section to explain why this period was chosen in the analysis:*

*"The threshold of 100% of the investigated years is a more stringent criterion with respect to other studies, generally adopting a less stringent criterion (e.g. 75% is set in Colette et al. (2011), corresponding in same cases to 8 years). Our choice guarantees that the trend analysis is always based on an 8-year period, which can be considered quite robust. Indeed, several studies are available in literature, presenting trend analysis over similar or shorter periods (Zhai et al., 2019; Dufour et al., 2018; Sheng et al., 2018). Of course, data covering a longer period would strengthen our findings. Anyway, in this first study over Italy, the choice of the period to investigate was determined by the availability of coherent model results that have the same model setup for the years 2003 to 2010. More specifically, in the following years, AMS-MINNI simulations adopted a different setup (spatial domain, chemical mechanism, boundary conditions), that clearly affects time series homogeneity."*

Line 15: I suggest that throughout the manuscript the word "trend" is used instead of "slope" to avoid confusion.  I suggest that "slope" is only used in the section describing the trend methodology.

*We corrected and used trend instead of slope.*

Lines 29-34: This is a good concise summary of the impacts of air pollution

*Thank you.*

Line 106: The spatial representativeness of TRA and IND stations is probably a lot smaller than the model spatial resolution. Although it is always good idea to evaluate the models with all possible observations it should be made clear that it is not expected that the model can realistically simulate the concentrations well at these sites

*We have pointed out that the model spatial resolution is not expected to cover the spatial representativeness of TRA and IND stations adding the following comment:*

*"The model spatial resolution of 4 km is not sufficient to describe TRA and IND stations with the same skills of BKG stations, nevertheless for the sake of completeness we chose to include them in the validation."*

Line 145: From the text it sounds like the finest spatial disaggregation was done at NUTS3 but from the trends maps it looks like a finer spatial aggregation was applied (e.g. to road emissions). Details of this finer spatial disaggregation should be included

*The emissions at NUTS3 were disaggregated on our grid model resolution using spatial layers for diffuse/linear sources and geographical coordinates for point sources, so producing a gridded emission inventory with a resolution of 4 km that was later speciated and modulated in our emission pre-processing. The following sentence was added to include more details about the spatial disaggregation:*

*"The spatial allocation of NUTS3 emissions to the 4 km grid of the MINNI model relied for point sources on geographic coordinates of each facility (for example, large combustion plants) and for diffuse/linear sources on spatial layers used as proxy variables, like population density (for residential heating and urban traffic), georeferenced road networks (for rural and highway traffic), land-use (for agriculture)".*

Lines 161-168: I'm not sure if this paragraph is very useful. The previous paragraph says that no clear meteorological trends were found so maybe this paragraph could be removed

*This part of the paragraph has been eliminated.*

Line 198: I don't think that 3 indicators could be considered a comprehensive set

*We agree with the comment: we made a mistake in the final language revision. We have corrected the sentence to make clearer the description of our analysis flow. Indeed, a comprehensive set of statistical indices was actually computed but only a subset of them was chosen for the discussion. Since this paper was focused on trend analysis and not on model validation, a complete discussion*

*of all the computed statistical scores was out of the scope of this paper. So, a choice of a subset was made, according to some criteria, as it was explained in the following of the Section 3.1. Briefly, the reasons are:*

*i) RMSE, MB and corr do globally capture several features of model performance in terms of amplitude, phase and bias;*

*ii) these indicators are frequently used in model evaluation studies. Simon et al. (2012) studied the frequency of use of several statistical metrics in model evaluations and they said that: "The most commonly reported metrics are **mean bias**, normalized mean bias, normalized mean error, and r/r2. Both ME and **RMSE** are frequently used as non-normalized error metrics";*

*iii) the same set of three statistical indices (i.e. RMSE, MB, corr) are presented in Colette et al. (2011).*

Line 214: AMS-MINNI performs even better. Better than what? Better than RMSE or corr (and if so, how was it evaluated?) or better than Colette 2011?

*We were referring to Colette et al. (2011) but the sentence was not clear. The sentence was removed and the comparison with Colette et al. (2011) was moved below and made more explicit.*

Lines 215-217: This may be true for TRA and IND sites but you would not expect this to be a problem for BKG sites

*We agree. The sentence now refers to the worsening of the performances at TRA sites.*

Line 232: "similar RMSE values". Similar to what? Similar to Colette 2011 or something else?

*We were referring to Colette et al. (2011). An explicit reference to the new Table S2 was added in the sentence in order to make it clearer.*

Line 238: "better". Better than what? the values are quite similar to the daily indicator

*MDA8 correlation skills are better compared to the daily indicator, for all the station types:*
*BKG, rural:          daily: 0.757    MDA8: 0.817*
*BKG, suburban:       daily: 0.814    MDA8: 0.847*
*BKG, urban:          daily: 0.822    MDA8: 0.853*
*IND:                 daily:  0.683   MDA8: 0.712*
*TRA:                 daily: 0.761    MDA8: 0.801*
*Of course, the improvement is slight. The sentence was so rephrased according to the suggestion, and "globally better" was replaced by "generally better", weakening the strength of the sentence.*

Lines 270-273: I suggest rephrase this is follows: Figure 3 shows that all STs are negative (93% significant), whereas 79% of the OTs were negative (58% significant; 21% non-significant) and 21% were positive (7% significant; 14% non-significant). (Please check my figures because I've estimated them from the figure)

*We have rephrased the sentence according to your suggestion.*

Line 276: Suggest add "at BKG sites" after "values" because model performance does not look that good for TRA sites in Fig 5a

*We have added "at BKG sites" after "values".*

Lines 292-293: This improvement appears to be related to the increase in PM emissions for SNAP2 (maybe residential heating emissions). Does this mean that these emissions were underestimated before that year? This will obviously have an impact on the trend estimates. I think that the authors should comment on this.

*The improvement we estimated in PM10 modelled concentrations after the year 2007 is not directly linked to emission estimates. There is no emission underestimation before that year as we detailed in the response to the comment to Figure S6 and in the text of the manuscript. The PM10 emission time series shows an increase from 2005 to 2008 and a reduction afterwards while model results improve from 2007 onwards. There is no clear evidence on the reasons of the improvement of PM10 simulated monthly means.*

Lines 366-367: Although the model skill for the trends could be considered good it is concerning that there is quite a large bias in concentrations, especially at the beginning of the time series. I think this should be looked into for future evaluations and maybe commented on here

*We agree and we mentioned this point in the revised version of the manuscript.*

Line 389: What does "half summer period" mean?

*The definition "half summer period" was rather generic. The calculation of the ozone metric, SOMO0, was calculated in the report of Cattani et al. (2014) in the period Apr-Sep. In the revised manuscript we substituted "in the half summer period" with "from April to September".*

Lines 413-414: The phrase "For almost all Northern Italy and for all pollutants, it is possible to estimate an area with significant simulated trends." does not make sense. Surely you can estimate the area with significant trends for the entire domain!

*What we meant here and we would like to underline is that the modelled trend has statistically significant values especially in the Northern domain, and less in Central and Southern Italy although with a greater area of significant trend with respect to observations. To be clearer, we rephrased the whole sentence as following "For all pollutants, almost the entire domain of Northern Italy has significant simulated trends.".*

**Supplementary material**:

Figure S6: What is the reason for the increase after 2006 in SNAP2 (residential biomass burning?) Is it real or just an adjustment to the estimates?

*The increase in PM emissions from the SNAP2 sector is linked to the increase in wood burning from the residential sector as documented from the Eurostat energy balance (https://appsso.eurostat.ec.europa.eu/nui/submitViewTableAction.do), synthetized in the following table, and reported in the different version of the Informative Inventory report for the Italian Emission Inventory (IIR, 2021, http://www.sinanet.isprambiente.it/it/sia-ispra/serie-storiche-*

*emissioni/informative-inventory-report/view), as we described in the manuscript, so it is not an adjustment to the estimates.*

*The wood consumption time series for the years 2003-2010 shows an increase from 2005 to 2008 and a reduction onwards. This trend is reflected in PM2.5 emissions from the SNAP2 sector whose main emissions derive from the residential burning of biomass (fig. S6 of the Supplementary Material) and also in the overall time series of total PM emissions (as results also from fig. 1 of the manuscript).*

*Table AC1 – Primary solid biofuels consumption from the households (source Eurostat).*

| Year | Primary solid biofuels (PJ) |
|------|------------------------------|
| 2003 | 144.350 |
| 2004 | 92.195 |
| 2005 | 161.868 |
| 2006 | 193.058 |
| 2007 | 269.118 |
| 2008 | 320.421 |
| 2009 | 307.158 |
| 2010 | 297.801 |

*We added the reference to the IIR, 2021 where it is clearer that the increase observed in PM emissions from the SNAP2 after 2006 is real and due to an increase in wood consumption from the residential sector.*

S5: These tables are very useful and complete

*Thank you.*

**Technical corrections (and suggestions):**

Line 14: Suggest change "regional models applications" to "regional air quality modelling"

*Done.*

Line 16: Suggest that the division sign (÷) should be changed for a dash. A division sign does not make sense in this context

*Done.*

Line 16: Suggest change "show the same magnitude" to "were of a similar magnitude"

*Done.*

Line 16: Suggest change "while a smaller variability is detected" to "while a smaller range of trends was found than those observed"

*Done.*

Lines 18-19: Suggest change "allowed to extend both the spatial coverage and the statistical significance of pollutants' concentrations trends" to "provides a greater spatial coverage and statistical significance of pollutant concentration trends"

*Done.*

Line 22: Remove final "s" from "concentrations" and "emissions"

*Done.*

Line 23: Remove final "s" from "pollutants"

*Done.*

Line 36: Suggest change "bringing " to "leading"

*Done.*

Line 37: Suggest change "measured concentrations of air pollutants" to "air pollutant concentrations"

*Done.*

Line 38: Suggest change "how much" to "to what degree"

*Done.*

Line 38: Suggest change "limitations" to "limits" and remove final s and apostrophe from "pollutants'"

*Done.*

Line 44: Suggest add "sources of " before "information"

*Done.*

Line 46: Suggest change "even if with" to "although they have"

*Done.*

Line 48: Suggest change "On" to "For"

*Done.*

Line 55: Suggest change "it is missing" to "failed to reproduce"

*Done.*

Line 60: Suggest change "scale (Colette et al., 2017a), they" to "scale, Colette et al. (2017a)"

*Done.*

Line 74: Suggest add "the studies by" before "Cattani"

*Done.*

Line 77: Suggest change "prevalence" to "number"

*Done.*

Line 78: Add comma after "trends"

*Done.*

Line 85: Remove final s from "concentrations"

*Done.*

Line 86: Suggest change "compared to meteorological and anthropogenic emissions variability" to "compared with variations in meteorology and anthropogenic emissions"

*Done.*

Line 89: Suggest change "where" to "in which"

*Done.*

Line 106: Suggest change "divided by type" to "by station type"

*Done.*

Line 107: Suggest change "ones divided by zone" to "sites by zone type"

*Done.*

Line 120: Suggest delete "in input"

*Done.*

Line 120: Suggest change "RAMS, orographic" to "RAMS and orographic"

*Done.*

Line 125: Add space before "km" (Horizontal resolution of RAMS) and before "m" (Vertical resolution)

*Done.*

Line 153: Suggest change "tendency" to "trends"

*Done.*

Line 175: Suggest remove "before time trend estimates"

*Done.*

Line 177: Suggest remove "statistical"

*Done.*

Line 177: Suggest change "it does not care about" to "it is not sensitive to"

*Done.*

Line 177: Suggest change "works" to "studies"

*Done.*

Line 181: Suggest change "not significant" to "non-significant" throughout the manuscript

*Done.*

Line 181: Suggest change "simulations analyses" to "modelled estimates"

*Done.*

Line 183: Suggest remove "all the"

*Done.*

Line 183: Suggest change "divided in" to "separated into"

*Done.*

Line 188: Suggest change "AMS-MINNI capability" to "capability of AMS-MINNI"

*Done.*

Lines 193-194: Suggest change "being this metric" to "this metric being"

*Done.*

Lines 194-195: Suggest change "Anyway, in addition to daily values and concerning only O3 evaluation" to "For O3, in addition to daily values"

*Done.*

Line 201: I think "indices" would be more correct than "indexes"

*Done.*

Line 203: Suggest change "time trends" to "temporal trends" throughout the manuscript

*Done.*

Line 205: Suggest remove "being them"

*Done.*

Line 206: Suggest change "concerning" to "for"

*Done.*

Line 206: Suggest change "for zone" to "by zone type"

*Done.*

Line 212: Suggest change "in detail" to "specifically"

*Done.*

Line 221: Suggest change "as commonly in" to "which is common for"

*Done.*

Line 221: Suggest change "models. This is highlighted by" to "models, as shown by"

*Done.*

Line 222: Suggest change "Anyway, in this case the" to "However,"

*Done.*

Line 225: Suggest change "Saharan dust contribution" to "The contribution of Saharan dust"

*Done.*

Lines 226-228: Suggest rewrite as "As far as MB and corr are concerned, simulated PM10 concentrations are overall in agreement with observations, with values ranging from -12.8 to -3.9 µg m-3 and from 0.453 to 0.630, respectively."

*Done.*

Line 231: Suggest change "More in detail, considering results at" to "More specifically, for"

*Done.*

Line 232: Suggest change "skills" to "values"

*Done.*

Lines 232-233: Suggest change "Similarly to NO2 performance" to "Similarly to the performance for NO2"

*Done.*

Line 233: Suggest change "the MB in reproducing O3 levels changes" to "the MB of O3 concentrations changes"

*Done.*

Line 233: Suggest add "the" before "spatial"

*Done.*

Line 235: Suggest change "plays as O3 sink" to "acts as an O3 sink"

*Done.*

Line 237: Suggest change "to daily one" to "to  that for daily concentrations"

*Done.*

Line 238: Suggest change "globally" to "generally"

*Done.*

Lines 238-241: Suggest rewirte as "With respect to daily values, correlation for MDA8 (0.712 - 0.853) is globally better, as is MB (lower absolute values). With regards to RMSE (24.4 - 25.1 µg m-3) the values are worse at BKG stations and slightly better at IND and TRA sites."

*Done.*

Line 243: Suggest change "expected if compared to daily ones, due to" to "expected when compared with daily values for"

*Done.*

Line 246: Suggest change "turns out to perform quite well being the results" to "performs quite well, with the results being"

*Done.*

Lines 246-247: Suggest change "the state of the art of air quality model performances" to "the performances of state of the art of air quality models"

*Done.*

Line 247: Suggest change "concerning" to "when considering"

*Done.*

Lines 250-251: Suggest change "to make a comparison between observed trends (OT) and simulated trends (ST)"  "to compare observed trends (OT) and simulated trends (ST)"

*Done.*

Line 255: Suggest remove "synthetically"

*Done.*

Lines 257-258: Suggest change "the time series of observed and simulated monthly average concentration values, detailing model performance in the considered multiannual period" to  "the time series of observed and simulated monthly average concentrations (averaged over all stations for each station type)"

*Done.*

Line 260: Suggest change "the map of the simulated" to "maps of simulated"

*Done.*

Line 260: Suggest insert "the" before "spatial"

*Done.*

Lines 260-261: Suggest change "split up according to station type" to "by station type"

*Done.*

Line 264: Suggest remove "description"

*Done.*

Line 269: Suggest change "show" to "have"

*Done.*

Line 274: Suggest change "better reproduces monthly values" to "reproduces monthly values better"

*Done.*

Line 278: Suggest change "worsen" to "is worse"

*Done.*

Line 278: Suggest change "usually" to "mostly"

*Done.*

Line 280: Suggest change "also in some portion of the territory" "in parts of the domain"

*Done.*

Line 282: Suggest move "better" to after "slopes"

*Done.*

Line 282: Suggest change "performances decline" to "is worse"

*Done.*

Lines 282-284: Suggest rewrite as "The map of the simulated slopes not only has a wider coverage but also shows a greater area with significant trends compared with observations."

*Done.*

Line 287: Suggest change "proven also" to "shown"

*Done.*

Line 288: Suggest change "show" to "have"

*Done.*

Lines 289-290: Suggest move "statistically significant" to before "observed"

*Done.*

Lines 290-291: Suggest change "reaches a percentage of 84% but decreases to 44% when simulated data are taken into account" to "is 84% compared with only 44% for the ST"

*Done.*

Line 292: Suggest change "slightly improving" to "improving slightly"

*Done.*

Line 294: Suggest change "in many" to "at many"

*Done.*

Line 295: Suggest remove "slightly".  It is too subjective

*Done.*

Line 296: Suggest remove "represented"

*Done.*

Line 296: Suggest change "depicting" to "", which show"

*Done.*

Line 297: Suggest change "if compared to" to "when compared with"

*Done.*

Line 297: Suggest change "also for PM10 the simulated statistically significant trends" to "the simulated statistically significant trends for PM10"

*Done.*

Lines 298-299: I don't understand the statement of poor coverage by the model when the model covers the entire country. I assume this should say observations.

*What we meant with those sentence is related to poor coverage of significant modelled trend although the model elaborates trends for the entire country. We rephrased the sentence as following "A poor coverage of significant trends in both model and observations occurs in some areas of Central and Southern Italy".*

Line 299: Should "Centre" not be "Central"?

*Done.*

Line 299: Suggest change "where anyway the model shows" to "where the model estimates"

*Done.*

Line 300: Suggest change "ST" to "trends"

*Done.*

Line 306: Suggest change "Apr/Sep" to "Apr-Sep" throughout manuscript

*Done.*

Line 308: Suggest change "on" to "for"

*Done.*

Lines 308-309: Suggest rewrite as "The number of stations with increasing and decreasing trends and their significance depends on the metric used (Table 2 and Fig. 11)."

*Done.*

Line 309: Change "trend" to "trends"

*Done.*

Line 309: Suggest change "observed" to "the observed"

*Done.*

Line 310: Suggest change first "trends" to "datasets"

*Done.*

Line 310: Suggest change "show" to "have"

*Done.*

Line 314: Suggest change "shows" to "has"

*Done.*

Line 319: Suggest change  "in observed trend than simulated ones." to "for observed trends than for those simulated."

*Done.*

Line 320: Suggest change "a wide area of significant simulated trend" to "a large area of significant simulated trends"

*Done.*

Line 321: Suggest rewrite "with no significant ST covering especially the North-Eastern area." as "with an area of non-significant ST in the North-Eastern area"

*Done.*

Line 322: Suggest change "where a higher sample is available" to "for which there are more stations available"

*Done.*

Line 322: Suggest change "the simulated trend" to "the model"

*Done.*

Line 323: Suggest change "observed trend where positive" to "observed positive trends"

*Done.*

Lines 323-324: Suggest rewrite "when significant decreasing OT are calculated the model shows a good agreement, even though with a lower variability range than observations." as "the model has a good agreement with the significant decreasing OT, although with a lower variability"

*Done.*

Lines 324-325: Suggest rewrite "are different areas, especially in the" to "are some areas, especially in"

*Done.*

Line 326: Suggest change "or without significant OT" to "or the OT is not significant"

*Done.*

Lines 328-329: Suggest rewrite as "Our analysis shows that AMS-MINNI is capable of reproducing observed trends albeit with some differences between the pollutants studied."

*Done.*

Lines 329-333: Suggest rewrite as "Although a quantitative analysis of the influence of variations in emissions and meteorology on concentration trends, we present a preliminary qualitative attempt to compare the temporal concentration trends to variation in emissions, having already observed (see Section 2.4) that there is no clear tendency in the meteorology."

*Done.*

Line 334: Suggest rewrite as "The nitrogen oxides (NOX) that are most relevant for air pollution (namely NO and NO2) are mostly emitted….."

*Done.*

Line 337: Suggest change "being" to "since"

*Done.*

Line 337: Suggest add "are" before "directly"

*Done.*

Line 339: Suggest remove "absolute values of"

*Done.*

Line 339: Suggest change "reproduces adequately" to "adequately reproduces"

*Done.*

Line 340: Add "a" before "national"

*Done.*

Line 340: Suggest change "showing robustness in potential support to" to "demonstrating its potential for supporting"

*Done.*

Line 342: Suggest change "decreasing tendency of concentrations measured" to "decreasing concentrations trends observed"

*Done.*

Line 343: Suggest change "or producing concentrations not fully responsive to correct trends" to "or the model is not responding correctly to the changes in emissions"

*Done.*

Lines 344-345: Suggest change "model capacity in capturing high gradients in concentration features" to "model's ability to capture large concentration gradients"

*Done.*

Line 345: Suggest change "model failure in catching positive slopes" to "failure of the model to capture the positive trends"

*Done.*

Line 346: Suggest change "exhibiting" to "with"

*Done.*

Line 347: Suggest change "by" to "in"

*Done.*

Line 348: Suggest change "points out" to "shows"

*Done.*

Line 349: Suggest change "sites exhibiting not significant OT" to "sites that have non-significant OT"

*Done.*

Line 351: Suggest change "simulated concentrations show no significant or decreasing trends" to "simulated trends are not significant or decreasing"

*Done.*

Line 351: Suggest change "like this happens in" to "similar occurs in"

*Done.*

Line 352: Suggest change "where positive slopes occur, being they obtained only from measurement analysis." to "with positive OT."

*Done.*

Line 353: Suggest change "are found" to "is observed"

*Done.*

Line 355: Suggest change "showing" to "with"

*Done.*

Lines 355-356: Suggest change "it is confirmed" to "these results suggest"

*Done.*

Line 358: Suggest remove "reductions"

*Done.*

Lines 359-360: Suggest change "chemical precursors (NOX, SOX, NH3, NMVOC) reactions." to "reactions of chemical precursors (NOX, SOX, NH3, NMVOC)."

*Done.*

Line 361: Suggest remove "quantitatively"

*Done.*

Line 361: Suggest change "on sites" to "on the site"

*Done.*

Line 364: Suggest change "of not significant trend illustrated in" to "of non-significant trends shown in"

*Done.*

Line 366: Suggest change "emission" to "emissions"

*Done.*

Line 369: Suggest change "on" to "for"

*Done.*

Line 370: Suggest remove "totally"

*Done.*

Line 373: Suggest change "emission and concentration time trends" to " temporal trends of emissions and concentrations"

*Done.*

Line 373: Suggest change "complex generation chain" to "complex photochemistry"

*Done.*

Lines 375-376: Suggest rewrite as "This is particularly important in Mediterranean áreas, which are susceptible to ozone-related impacts (De Marco et al., 2019) due to climatological conditions that are more favourable for O3 formation"

*Done.*

Line 377: Suggest change "gradient" to "trend"

*Done.*

Lines 377-379: Suggest rewrite "considered, so not modifying the national scale ratio between the two, which is the main driver of the chemical equilibrium in the generation of O3 (Seinfeld and Pandis, 1998; Sillman, 1999)." to "considered, thus there has been little change in the ratio between them, which is the main driver of the chemical equilibrium for O3 formation (Seinfeld and Pandis, 1998; Sillman, 1999).

*Done.*

Line 380: Suggest change "spot densely populated with" to "spot, densely populated and with"

*Done.*

Line 382: Suggest change "in some scattered stations:" to "for some stations:"

*The sentence was changed and this expression is not present any more.*

Line 384: Suggest change "precursors'" to "precursor"

*Done.*

Line 385: Suggest change "could be additional causes of" to "could lead to"

*Done.*

Line 385: Suggest change "were already noticed in" to "can be found in the"

*Done.*

Line 387: Suggest change "all over" to "throughout"

*Done.*

Line 388: Suggest change "even if on" to "although with"

*Done.*

Line 390: Suggest change "among" to "between"

*Done.*

Line 391: Suggest remove "the" before "AMS"

*Done.*

Line 393: Suggest change "values" to "estimates"

*Done.*

Line 395: Suggest change "in" to "of"

*Done.*

Line 396: Suggest change "on" to "for"

*Done.*

Line 398: Suggest add "the" before "literature"

*Done.*

Line 399: Suggest change "of regional models applications" to "art for regional model applications"

*Done.*

Line 399: Suggest change "turned out to be" to "are"

*Done.*

Line 400: Suggest change "and the same behaviour of most of the regional models" to "and a similar behaviour to that of most regional models"

*Done.*

Line 404: Suggest remove "out"

*Done.*

Lines 407-407: Suggest change "Concerning PM10, the reason for this discrepancy" to "The reason for the discrepancy for PM10"

*Done.*

Line 407: Suggest change "PM10 modelled" to "modelled PM10"

*Done.*

Line 408: Suggest change "O3 results" to "The results for O3"

*Done.*

Line 408: Suggest add "data" after "network"

*Done.*

Line 409: Suggest change "catching" to "capturing"

*Done.*

Line 411: Suggest change "variability description were illustrated" to "variability are illustrated"

*Done.*

Line 411: Suggest change "maps generated for all" to "maps for"

*Done.*

Line 412: Suggest change "wider significant area for" to "larger area for significant"

*Done.*

Line 412: Suggest change "to observed ones" to "with these observed"

*Done.*

Line 412: Suggest change "higher" with "larger"

*Done.*

Line 413: Suggest change "lower" for "smaller"

*Done.*

Lines 414-415: Suggest change "PM10 significant modelled trends" to "significant modelled PM10 trends"

*Done.*

Line 415: Suggest change "it is possible to catch some" to "there are"

*Done.*

Line 415: Suggest change "simulated significant trends" to "significant simulated trends"

*Done.*

Lines 415-416: Suggest change "where no observed data are present at all." to "where there are no observations."

*Done.*

Line 416: Suggest change "More in detail, it is worth" to "It is also worth"

*Done.*

Line 417: Suggest change "due to the absence of significant observed trends" to "due to a sparse or absent monitoring network"

*Done.*

Line 419: Suggest change "meteorology" to "meteorological"

*Done.*

Line 420: Suggest remove last s from "emissions"

*Done.*

Line 421: Suggest remove last s from pollutants

*Done.*

Line 425: Suggest change "widening" to "increasing"

*Done.*

Line 426: Suggest change "turned out to be better for" to "is best"

*Done.*

Line 475: Suggest change "elaborations" to "calculations"

*Done.*

Line 806: Suggest remove "as a percentage" (Figure 1)

*Done.*

Line 806: Suggest move "from 2003 to 2010" to before "relative"

*Done.*

Line 814: Suggest change "in the period" to "for the period" (Figure 2)

*Done.*

Lines 817-818: Suggest change both instances of "dashed" to "hashed" (Figure 3 and for all similar figures)

*Done.*

Line 822: Suggest change "dash" to "dashed" (Figure 4 and for all similar figures)

*Done.*

Line 823: Suggest change "stations" to "station" (and in all similar figures)

*Done.*

Line 828: Suggest change "lilac square" to "pink square" since I think pink is more well known than lilac

*Done.*

Lines 861-862: Suggest change "for different O3 observed and simulated metrics" to "for different observed and simulated metrics for O3"

*Done.*

**Supplementary material**:

Figure S5: Suggest change "Emission trend" to "Emission time series" since these are not trends (and for figures S6 and S7)

*Done.*

Figure S8: Suggest add "Annual" before "Temperature" (also applies to subsequent figures)

*Done.*

S4 (p10): Suggest change "are reported in the" to "calculated according to the"

*Done.*

S5.1: Suggest change "all over the Italian territory." to "for the entire Italian territory."  (also applies to the subsequent tables)

*Done.*